# WHAT MAKES A GOOD DIFFUSION PLANNER FOR DECISION MAKING?

**Haofei Lu**[1]  **Dongqi Han**[2][†]  **Yifei Shen**[2]  **Dongsheng Li**[2]
[1]Tsinghua University    [2]Microsoft Research Asia
luhf23@mails.tsinghua.edu.cn
{dongqihan,yifeishen,Dongsheng.Li}@microsoft.com

## ABSTRACT

Diffusion models have recently shown significant potential in solving decision-making problems, particularly in generating behavior plans – also known as diffusion planning. While numerous studies have demonstrated the impressive performance of diffusion planning, the mechanisms behind the key components of a good diffusion planner remain unclear and the design choices are highly inconsistent in existing studies. In this work, we address this issue through systematic empirical experiments on diffusion planning in an offline reinforcement learning (RL) setting, providing practical insights into the essential components of diffusion planning. We trained and evaluated over 6,000 diffusion models, identifying the critical components such as guided sampling, network architecture, action generation and planning strategy. We revealed that some design choices opposite to the common practice in previous work in diffusion planning actually lead to better performance, e.g., unconditional sampling with selection can be better than guided sampling and Transformer outperforms U-Net as denoising network. Based on these insights, we suggest a simple yet strong diffusion planning baseline that achieves state-of-the-art results on standard offline RL benchmarks.

## 1 INTRODUCTION

Decision making by learning from offline data has been a fundamental approach in robotics and artificial intelligence (Bellman, 1957). It enables agents to acquire complex behaviors by observing and mimicking expert demonstrations, circumventing the need for explicit programming or exhaustive exploration. However, this paradigm faces significant challenges, particularly when dealing with long-horizon planning and high-dimensional action spaces. The complexity of modeling sequential dependencies and capturing the intricacies of action distributions makes it difficult to scale traditional methods (Deisenroth & Rasmussen, 2011) to more complex tasks (Parmas et al., 2018).

Recently, diffusion models have achieved remarkable success in image and video generation, demonstrating their ability to handle complex distribution and long-range dependencies (Ho et al., 2020; Dhariwal & Nichol, 2021). Inspired by these works, several recent studies have applied diffusion models to planning sequential decisions, especially with continuous state and action spaces such as robotic manipulation tasks (Janner et al., 2022; Ajay et al., 2022; Lu et al., 2023; Li et al., 2023). The diffusion models are used to approximate the sequence of states and actions from current time step to future – and by exploiting the diffusion models' conditional generation capacity such as diffusion guidance (Ho et al., 2020; Ho & Salimans, 2021), the model can make plans (i.e. state trajectory) with desired properties such as reward maximization (i.e. offline reinforcement learning (Levine et al., 2020)).

Despite achieving impressive performance across a diverse array of tasks, there has been limited exploration into the fundamental components and mechanisms that constitute an effective diffusion planning model for decision making. Previous research exhibits a lack of consistency and coherence in design choices. It remains uncertain whether sub-optimal design choices might hinder the full

---

This work was done during the internship of Haofei Lu (luhf23@mails.tsinghua.edu.cn) at Microsoft Research Asia. Correspondence to: Dongqi Han <dongqihan@microsoft.com>

potential of diffusion models within decision-making domains. Specifically, existing approaches have not adequately addressed essential facets such as the choice of diffusion guidance algorithm, network architecture, and whether the plan should contain states or state-action pairs. This naturally raises the following fundamental question:

*What makes a good diffusion planner for decision making, especially offline RL?*

We seek to answer the question by conducting a comprehensive empirical investigation into key design choices in diffusion models for decision-making, in particular for state-based robotics tasks. Our work contributes to the field of decision making and diffusion models in several aspects.

- **Comprehensive experiments:** We conducted an extensive empirical study to explore what constitutes an effective diffusion planner. By training and evaluating over 6,000 models, we analyzed key components critical to decision making in diffusion planning, including guided sampling algorithms, network architectures, action generation methods, and planning strategies.
- **Insights and tips:** We ran detailed experiments and data analysis to understand the role of each key component in constituting a good diffusion planner. In particular, we discovered that certain design choices, contrary to common practice in diffusion planning actually lead to better performance. Our work offers intuitive explanations and practical tips about the choices and provides insights about the strengths and limitations of diffusion planning.
- **A simple yet strong baseline:** Building on the insights from our study, we suggest a simple yet highly competitive baseline, named *Diffusion Veteran* (DV). This model achieves state-of-the-art performance in planning tasks in standard offline RL benchmarks.

## 2 BACKGROUND AND RELATED WORK

**Offline Reinforcement Learning** (Fujimoto et al., 2019; Levine et al., 2020; Fu et al., 2020) is a subfield of reinforcement learning (RL) where the agent learns from a fixed dataset of past experiences. This dataset typically consists of state-action-reward-next-state tuples, which encapsulate the agent's interactions with the environment. The challenge in offline RL is for the agent to derive an effective policy from this static dataset without further exploration or interaction with the environment. Two major challenges arise in this context. First, the state and action spaces may be high-dimensional and involve long-range dependencies, making it difficult to model effectively (Levine et al., 2020). Second, the learned policy must be optimal, even though the behavior policy that generated the offline data may be sub-optimal or different from the desired policy (Fujimoto et al., 2019).

Recently, diffusion models have emerged as a powerful framework for tasks such as image and video generation due to their ability to model complex distributions (Croitoru et al., 2023), which could mitigate the first problem. Moreover, diffusion guidance techniques (Ho et al., 2020; Ho & Salimans, 2021) allow the model to generate samples that adhere to the desired properties. The second challenge in offline RL, learning an optimal policy, can be addressed by diffusion guidance techniques to produce behavior that maximizes rewards. Building on this insight, a growing body of research has explored the use of diffusion models to generate behavior trajectories, denoted as $\tau$.

**Diffusion planning** (Ajay et al., 2022; Janner et al., 2022; Liang et al., 2023; Dai et al., 2023; Yang et al., 2023; Li et al., 2023; Yang et al., 2023; Chen et al., 2024; Dong et al., 2024c) considers that at the time step $t$, a *trajectory* $\tau$ consists of the current and subsequent $H$ steps of state-action pairs or states:

$$\tau = \begin{bmatrix} s_t & s_{t+1} & \cdots & s_{t+H-1} \\ a_t & a_{t+1} & \cdots & a_{t+H-1} \end{bmatrix}, \text{ or } \tau = \begin{bmatrix} s_t & s_{t+1} & \cdots & s_{t+H-1} \end{bmatrix}. \tag{2.1}$$

There is a guidance function to model the reward, such as the immediate reward $r_t$ or the state value function $v(s_t) = \mathbb{E}\left[\sum_{h=0}^{\text{end}} \gamma^h r_{t+h}\right]$, where $\gamma$ is the discount factor (Sutton & Barto, 1998). In classifier guidance (CG) (Ho et al., 2020), a guidance network is learned simultaneously with the diffusion model, whose input is the generated trajectory and the output is accumulated rewards or value function. The gradient of the guidance network is used in the generation process of diffusion model to maximize the rewards. Examples of diffusion planning with CG are (Janner et al., 2022; Liang et al., 2023; Zhang et al., 2022). In classifier-free guidance (CFG) (Ho & Salimans, 2021), it

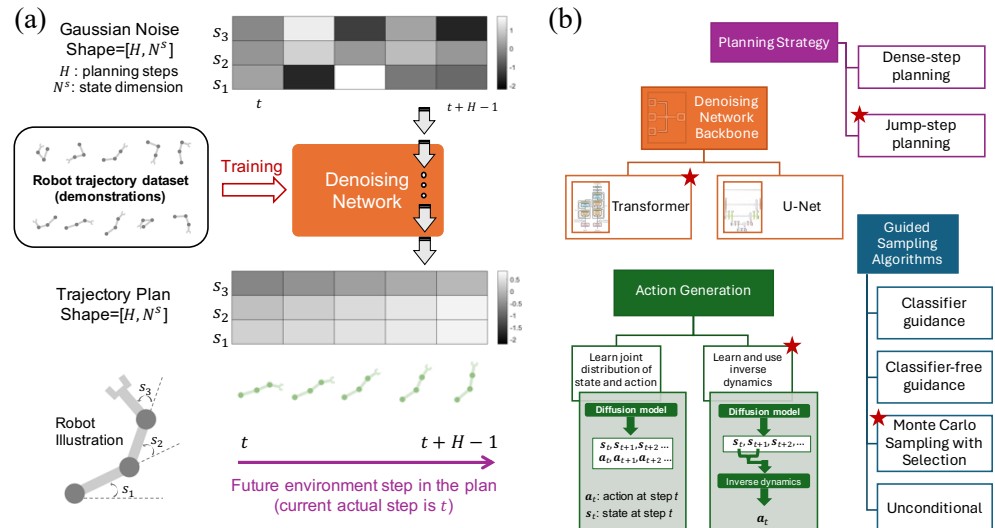

Figure 1: **Diffusion planning framework for decision making.** (**a**) The generation of a sequence plan using the denoising process of a diffusion model. A 3-joints robot arm is used as an illustrative example. (**b**) Important components and candidates in the framework. Each color corresponds to one component in the framework. A star indicates the preferred choice in experiments.

takes the desired reward or value function as an additional argument feed into the diffusion process. Instances are (Ajay et al., 2022; Li et al., 2023; Yang et al., 2023). However, despite some literature reviews such as Zhu et al. (2023), **the field lack a systematical study to elucidate the design space of diffusion planning in offline RL with substantial experimental results.**

**Diffusion policy** (Pearce et al., 2023; Wang et al., 2023b; Hansen-Estruch et al., 2023; Chen et al., 2023) is another kind of popular usage of diffusion model in decision making. The trajectory only includes $\tau = a_t$, without lookahead planning. The model is trained by combining the loss of imitation learning and model-free RL as in classic offline RL methods (Kumar et al., 2020; Fujimoto & Gu, 2021). Diffusion policy methods hope to improve the performance of by leveraging the capacity of diffusion model to model complex distribution of actions (policy function). A recent study (Dong et al., 2024b) investigated the design space of diffusion policy, proposed that diffusion policies such as DQL (Wang et al., 2023b) can be a computationally efficient and powerful candidate for decision-making tasks.

# 3 STUDY DESIGN

## 3.1 KEY COMPONENTS AND MECHANISMS OF DIFFUSION PLANNER

Recent pioneering work in diffusion planning (Janner et al., 2022; Ajay et al., 2022; Chen et al., 2024) has demonstrated the potential of this approach in offline RL. However, the design choices in these studies vary significantly, and it remains unclear whether there is an optimal configuration for different domains. Our aim is to conduct a systematic analysis supported by comprehensive experimental results. To achieve this, we begin by listing key design components (excluding common deep learning hyperparameters such as learning rates) that have varied in previous studies. See Fig. 1(b) for an overview.

**Guided sampling algorithms:** Classifier guidance (CG) (Ho et al., 2020), Classifier-free guidance (CFG) (Ho & Salimans, 2021), Monte Carlo sampling with selection (sample N *unconditional* trajectories and select the best, the criteria of which is given by a critic function learned simultaneously with diffusion model). Most previous diffusion planners used CG (Janner et al., 2022; Wang et al., 2023a; Chen et al., 2024) or CFG (Ajay et al., 2022; Li et al., 2023; Yang et al., 2023) for offline RL.

**Denoising network backbone:** U-Net (Ronneberger et al., 2015); Transformer (Vaswani et al., 2017). U-Net was used in most previous diffusion planners for state-based offline RL (Janner et al., 2022; Ajay et al., 2022; Wang et al., 2023a; Li et al., 2023; Chen et al., 2024)).

**Action Generation:** Learn joint distribution of state and action and directly execute the generated action at the current step (used in, e.g. Janner et al. (2022); Liang et al. (2023)); Learn and use inverse dynamics to compute action from state plan (used in e.g., Ajay et al. (2022); Wang et al. (2023a)).

**Planning strategy:** Dense-step planning means the planned trajectory $\tau$ (Eq. 2.1) corresponds to contiguous H steps in the environment (this is a conventional setting in diffusion planning (Janner et al., 2022; Ajay et al., 2022; Lu et al., 2023)) ; Jump-step planning models $H \times m$ environment steps, where $m \in \mathbb{N}^+$ is the planning stride; Hierarchical planning (studied by Li et al. (2023); Chen et al. (2024)).

Details of the implementation are deferred to Appendix A and B.

## 3.2 EXPERIMENT PROCEDURE

Given the multitude of components involved, it is challenging to draw scientific conclusions directly from the collective results. Therefore, we structured our study using the following procedure:
**(1)** Conduct a comprehensive search on the key components (Sect. 3.1) by combining grid search and manual tuning to obtain the best results.
**(2)** Evaluate the effect of each component using the control variable method; that is, modify only one component of the best model at a time and compare it with the original.
**(3)** After identifying which components are important and understanding how they affect performance, perform a deeper analysis to derive useful insights.

## 3.3 BENCHMARK

We conducted experiments on the D4RL dataset (Fu et al., 2020), one of the most widely used benchmarks for offline RL and imitation learning. The dataset covers a variety of task domains, including maze navigation, robot locomotion, robot arm manipulation, and vehicle driving, among others. For our experiments, we selected three sets of behavior planning tasks that were most commonly studied in prior works in offline RL and diffusion planning (Janner et al., 2021; Ajay et al., 2022; Janner et al., 2022; Liang et al., 2023; Li et al., 2023; Lu et al., 2023; Chen et al., 2024). These tasks (Fig. 2) encompass both planning and control challenges, providing a comprehensive evaluation in various problem settings. The performance metric considered in this work is the standard RL objective: the average total rewards in an online testing episode.

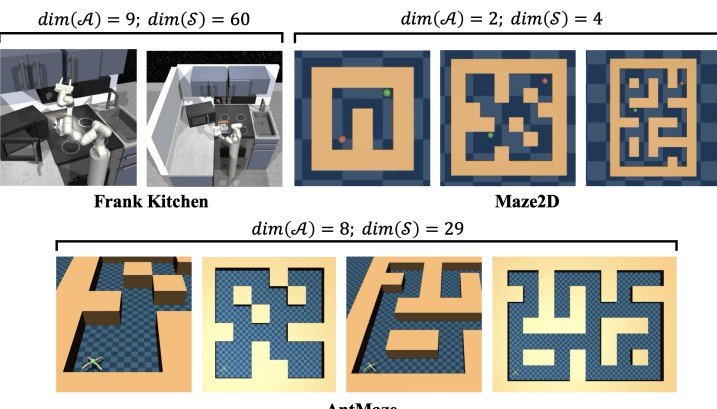

Figure 2: **Rendering of the benchmarking tasks considered in this study, where** $dim(\mathcal{S})$ **and** $dim(\mathcal{A})$ **denote the dimension of the state and action spaces.**

**Maze2D** involve navigating a 2D maze, requiring the agent to find an optimal path to a goal. These tasks are used to test planning capabilities in environments where spatial reasoning is critical.

**AntMaze** presents a navigation challenge with a simulated ant robot. The agent controls a multi-legged robot to navigate through a 2D maze, combining both locomotion and planning.

| Category | Env Dataset | Kitchen | | | Antmaze | | | | | Maze2D | | | |
|---|---|---|---|---|---|---|---|---|---|---|---|---|---|
| | | Mixed | Partial | avg. | L.-div. | L.-play | M.-div. | M.-play | avg. | L. | M. | Umaze | avg. |
| Non-diffusion | BC | 47.5 | 33.8 | 40.7 | 0.0 | 0.0 | 0.0 | 0.0 | 0.0 | 5 | 30.3 | 3.8 | 13.0 |
| | BCQ | 8.1 | 18.9 | 13.5 | 2.2 | 6.7 | 0.0 | 0.0 | 2.2 | 6.2 | 8.3 | 12.8 | 9.1 |
| | CQL | 51.0 | 49.8 | 50.4 | 61.2 | 53.7 | 15.8 | 14.9 | 36.4 | 12.5 | 5.0 | 5.7 | 7.7 |
| | IQL | 51.0 | 46.3 | 48.7 | 47.5 | 39.6 | 70.0 | 71.2 | 57.1 | 58.6 | 34.9 | 47.4 | 47.0 |
| Diffusion Policies | SfBC | 45.4 | 47.9 | 46.7 | 45.5 | 59.3 | 82.0 | 81.3 | 67.0 | 74.4 | 73.8 | 73.9 | 74.0 |
| | DQL | 62.6 | 60.5 | 61.6 | 56.6 | 46.4 | 78.6 | 76.6 | 64.6 | – | – | – | – |
| | DQL* | 55.1 | 65.5 | 60.3 | 70.6 | 81.3 | 82.6 | 87.3 | 80.5 | 186.8 | 152.0 | 140.6 | 159.8 |
| | IDQL | 66.5 | 66.7 | 66.6 | 67.9 | 63.5 | 84.8 | 84.5 | 75.2 | 90.1 | 89.5 | 57.9 | 79.2 |
| | IDQL* | 66.5 | 66.7 | 66.6 | 40.0 | 48.7 | 83.3 | 67.3 | 59.8 | – | – | – | |
| | CEP | – | – | – | 64.8 | 66.6 | 83.8 | 83.6 | 74.7 | – | – | – | |
| Diffusion Planners | Diffuser | 52.5 | 55.7 | 54.1 | 27.3 | 17.3 | 2.0 | 6.7 | 13.3 | 123 | 121.5 | 113.9 | 119.5 |
| | AdpDfsr | 51.8 | 55.5 | 53.7 | 8.7 | 5.3 | 6.0 | 12.0 | 8.0 | 167.9 | 129.9 | 135.1 | 144.3 |
| | DD | 75.0 | 56.5 | 65.8 | 0.0 | 0.0 | 4.0 | 8.0 | 3.0 | – | – | – | |
| | HD | 71.7 | 73.3 | 72.5 | 83.6 | – | 88.7 | – | – | 128.4 | 135.6 | 155.8 | 139.9 |
| | DV (Ours) | 73.6 | 94.0 | **83.8** | 80.0 | 76.4 | 87.4 | 89.0 | **83.2** | 203.6 | 150.7 | 136.6 | **163.6** |

Table 1: **Normalized performance of various offline-RL methods.** Our results (DV) are averaged over 500 episode seeds. The results of other methods are obtained from literature. We omit the variance over seeds for simplicity; however, it can be found in the detailed tables in Appendix D. The best average performance on each task set are marked in bold fonts. BC: vanilla imitation learning, BCQ: Fujimoto et al. (2019), CQL: Kumar et al. (2020), IQL: Kostrikov et al. (2021), SfBC: Chen et al. (2023), DQL: Wang et al. (2023b), IDQL: Hansen-Estruch et al. (2023), DQL* and IDQL*: replicated by Dong et al. (2024b), CEP: Lu et al. (2023), Diffuser: Janner et al. (2022), AdptDfsr: Liang et al. (2023), DD: Ajay et al. (2022), HD: Chen et al. (2024).

**Franka Kitchen** simulates a robot arm performing a variety of manipulation tasks in a kitchen environment to achieve task goals across multiple stages.

## 4 EXPERIMENTAL RESULTS

We trained and evaluated over 6,000 diffusion models by sweeping the key components discussed in Sect. 3.1 and other hyper-parameters (See Appendix B for details).

By summarizing the results from the experiments, we identified one kind of diffusion planning framework, called the Diffusion Veteran (DV). The pseudocode of DV can be found in Algorithm 1. As shown in Table 1, DV outperforms all previous diffusion planning and diffusion policy methods. We hope DV will serve as a simple yet strong baseline for future research in diffusion planning.

---

**Algorithm 1:** Diffusion Veteran (DV) Simplified Pseudocode

**Input:** Planning horizon $H$, Dataset $\mathcal{D}$, Discount factor $\gamma$, Candidate num $N$, Planning stride M.
**Initialize:** Diffusion Transformer Planner $\epsilon_\theta$, Diffusion Inverse dynamics $\epsilon_\omega$, Critic $V_\phi$

1  Calculate accumulated discounted returns $R_t = \sum_{h=0}^{\text{end}} \gamma^h r_{t+h}$ for every step $t$.
2  **Function** TRAINING:
3      Sample $\boldsymbol{s}_{t,t+M,\cdots,t+(H-1)M}, \boldsymbol{a}_{t,t+M,\cdots,t+(H-1)M}, R_t$ from $\mathcal{D}$
4      Train planner $\epsilon_\theta$ using $\boldsymbol{s}_t$ as condition and $\boldsymbol{s}_{t,t+M,\cdots,t+(H-1)M}$ as target output
5      Train Inverse dynamics $\epsilon_\omega$ using $\boldsymbol{s}_t, \boldsymbol{s}_{t+M}$ as input, $\boldsymbol{a}_t$ as target output
6      Train critic $V_\phi$ using $\boldsymbol{s}_{t,t+M,\cdots,t+(H-1)M}$ as input, $R_t$ as target output
7  **end**
8  **Function** EXECUTION(*s*):
9      Randomly generate $N$ plans using $\epsilon_\theta$, while fixing the first state as $\boldsymbol{s}$ during sampling
10     Select the best plan using critic $V_\phi$
11     Use the inverse dynamics $\epsilon_\omega$ to generate action using $\boldsymbol{s}$ and the next state in the best plan
12  **end**

---

With DV in place, we can then analyze the impact of each component in diffusion planning by looking into how each component influences its performance. Each of the following sub-sections will focus on one component that we have found to be crucial. In the end of this section, we will conclude our findings into practical tips.

## 4.1 ACTION GENERATION

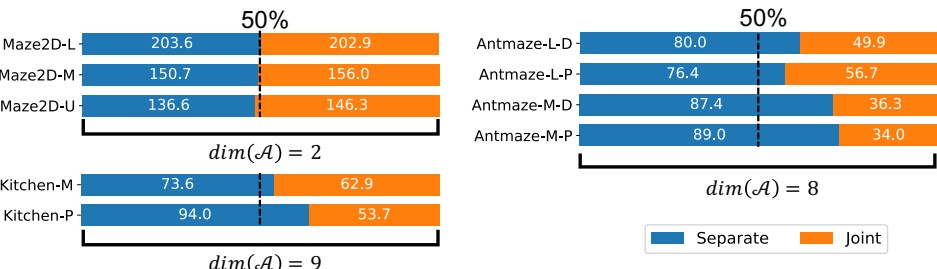

Figure 3: **Comparison of performance between two action generation strategies.** "Seperate" learns and uses inverse dynamics to compute action from state plan. "Joint" means learning joint distribution of state and action and directly executing the generated action at the current step (see "action generatation" in Fig. 1(b)). A straightforward conclusion drawn from the results is that "Separate" is better than "Joint" when tackling higher-dimensional action spaces. The vertical dashed line indicates on-par performance.

The choice of action generation design (Sect. 3.1) remains a subject of ongoing debate within the field. On one side, the pioneering diffusion planner, Diffuser, along with subsequent studies (Janner et al., 2022; Liang et al., 2023; Chen et al., 2024), employs a diffusion model to generate the joint distribution of action and state trajectories ("joint"). In contrast, studies by Ajay et al. (2022); Wang et al. (2023a); Du et al. (2024) have adopted inverse dynamics to generate actions based on planned states ("separate").

Our experimental findings favor the latter approach: Although both strategies perform comparably in simpler environments such as Maze2D, which lacks robotic control elements, the "separate" approach significantly outperforms the "joint" strategy in more complex settings like Kitchen and AntMaze, which feature robotic control and higher-dimensional action spaces.

This observed disparity may be attributed to the additional complexity introduced when modeling the joint distribution of sequential states and actions, compared to modeling only the states. This complexity becomes particularly pronounced in environments where state transitions involve more complex actions due to higher-dimensional action spaces.

We tested both diffusion models and vanilla MLP as the inverse dynamics, and found similar performance between them. We adhered to diffusion inverse dynamics (Appendix B.1).

## 4.2 PLANNING STRATEGY

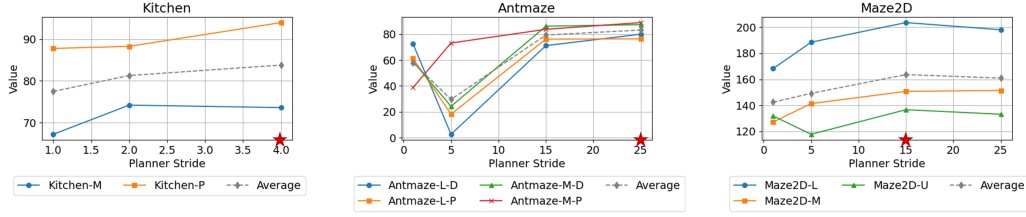

Figure 4: **Performance change of DV over planning stride**. It reduces to dense-step planning when Stride=1. The star indicates the choice of DV.

One crucial result we found is that jump-step planning (Sect. 3.1) is beneficial in almost all cases, despite the fact that most previous work used dense-step planning. This is observed in DV (Fig. 4) and generally in diffusion planners (see Appendix D for extensive results).

An obvious benefit from jump-step planning is that with the same planning steps, the model can look ahead farther. This may be crucial for planning tasks that require long-term credit assignment. The choice of stride should be related to the actual clock-time interval between two environment

steps. Nonetheless, we suggest to try jump-steps and sweep the stride. This observed phenomenon also implies that the diffusion planner should play the role of planning at a more abstract level or with a longer timescale. Interestingly, this is consistent with the neuroscientific fact that the intrinsic timescale of the prefrontal cortex (higher-level planning) is longer than that of the motor cortex (low-level control) (Murray et al., 2014; Runyan et al., 2017; Wang et al., 2018). A recent study (Chen et al., 2024) demonstrated impressive planning performance (Table 1, HD) using multi-timescale diffusion planning. Exploring the hierarchical paradigm of diffusion planning could be an interesting future direction.

## 4.3 DENOISING NETWORK BACKBONE

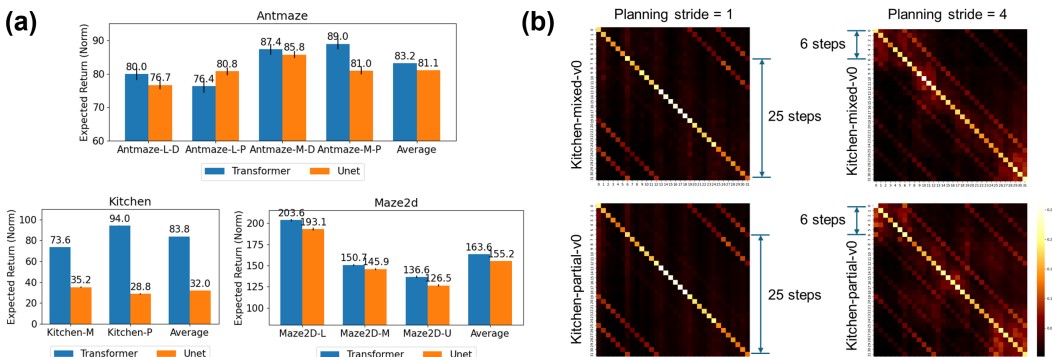

Figure 5: Using Transformer as the backbone of denoising network. (**a**) Performance comparison between Transformer and U-Net.The Transformer outperforms U-Net in 8 out of 9 sub-tasks and in all 3 main tasks. The amount of parameters in U-Net is comparable to that in Transformers. Note that the error bars in Kitchen are too small to visualize (See Table 10 for numerical results). (**b**) Visualization of attention weights of the first layer in the Transformer network during the denoising process. More plots can be found in Appendix D.

Most diffusion planners on the D4RL dataset use 1-D U-Net for the denoising network. It is natural to question whether attention is all you need (Vaswani et al., 2017) for diffusion planning. Thus, we examined the benefit of replacing U-Net with the Transformer architecture as the backbone of the denoising model (Sect. 3.1) (see Appendix B for details about network structures). The experimental results clearly support the utilization of Transformer (Fig. 5(a)) in diffusion planning, consistent with the latest trend in image and video generation (Peebles & Xie, 2023; OpenAI, 2024).

We conducted a case study by looking into the attention weights of the trained Transformer in the Kitchen environment (Fig. 5(b)), which reflect the temporal credit assignment (i.e., to how many steps later should be paid attention in the planning sequence). First, we see that the model pays more attention to the long-range element in the trajectory compared to the short-range ones. It suggests that the long-term dependency is crucial in this task, which breaks the local inductive bias of convolutional neural networks such as U-Net. Second, an interesting finding is that the characteristic attention length is consistent even with different planning stride (Sect. 4.2): 6 (attention step) × 4 (stride) ≈ 25 (attention step) × 1 (stride), as depicted in Fig. 5(b). It suggests that the Transformer finds the invariant correlations across the stride, contributing to the generalization performance.

More generally, we found long-term attention existing in the Transformer in the other tasks as well, although the attention patterns vary across different tasks.The attention patterns typically feature slashes, which attend to a fixed number of steps prior, and vertical lines, which attend to key steps. We have included the attention weights visualization in Appendix D. In-depth study will be needed to fully understand the role of long-term dependency and why Transformer is observed to outperform UNet in the future.

## 4.4 IMPACT OF NETWORK SIZE

Since the experimental results are in favor of Transformer, one may wonder whether a "scaling law" (Kaplan et al., 2020) holds, in particular, whether performance scales up with model depth (Ye et al.,

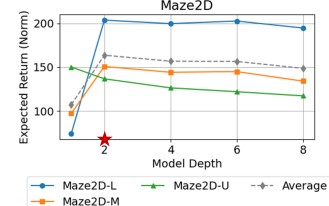

Figure 6: **Performance change over depth of the Transformer network as diffusion planner.** The star indicates the choice of DV.

2024). The results presented in Fig. 6 pass two clear messages: First, 1-layer Transformer is not enough, except for the simplist sub-task (Maze2D-U). Second, a deeper model is not always better. This may be due to a intrinsic difference between decision making and natural language processing and limitations of dataset size and quality, which requires further study to systematically address.

## 4.5 GUIDED SAMPLING ALGORITHMS

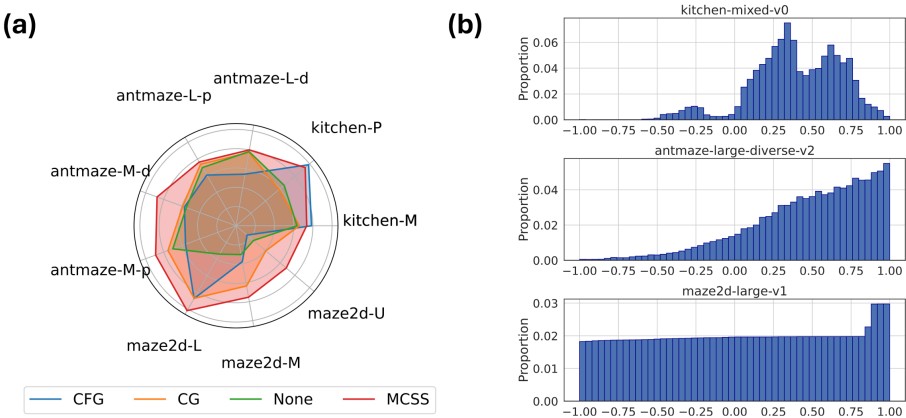

Figure 7: **Analysis of guided sampling algorithm.** (**a**) Performance comparison among different guided sampling algorithms for reward maximization. (**b**) Histogram of the value (accumulated discounted return in the future ($\sum_{h=0}^{\text{end}} \gamma^h r_{t+h}$, normalized to $[-1, 1]$) of the data points in each environment. For AntMaze, the failed trajectories are omitted since their values are all 0.

Another inconsistent design in previous work lies in the choice of guided sampling algorithm (Sect. 3.1), which enables the diffusion planner to generate plans that perform better than the average level of the dataset. Fig. 7(a) visualizes the corresponding empirical results (normalized) in our model. We can draw several conclusions from the results.

First, classifier guidance (CG) is comparable with classifier-free guidance (CFG), despite the fact that CFG is generally considered better than CG in image synthesis (Ho & Salimans, 2021). A potential reason is that the target value of CFG may need to be adjusted over time since the total rewards an agent can obtain in the future may vary depending on the task stage, but we can only use a fixed target value for CFG since there is no trivial solution.

Also, we observed that non-guidance can be better than guidance – Monte Carlo sampling with selection (MCSS) performs overall the best, except for Franka Kitchen where MCSS lags slightly behind CFG. This is an important finding since existing diffusion planners usually used CG or CFG (Chen et al., 2023; Wang et al., 2023b)). To understand the potential underlying reasons, we plotted the value distribution of data in each environment (Fig. 7(b)). It can be seen that in Maze2D and AntMaze, there is a substantial amount of optimal and near-optimal experiences, whereas in Kitchen most samples are sub-optimal (note that here the optimality is with respect to condition of diffusion model). This may explain why CFG performs better than MCSS in Kitchen. Thus we can propose a hypothesis: No guidance (MCSS) can be better than guided generation (CG, CFG) if the dataset contains a substantial portion of expert demonstration.

## 4.6 COMPARISON TO DIFFUSION POLICY

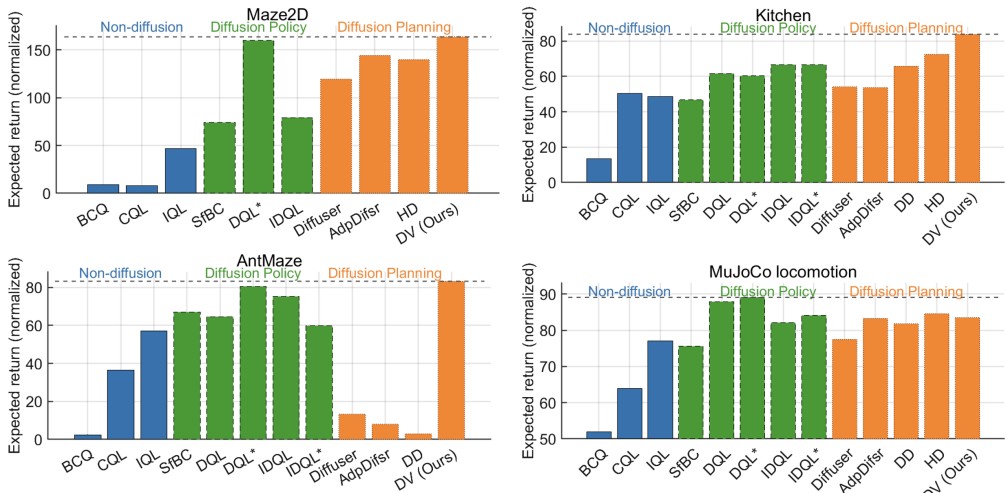

Figure 8: **Average performance of methods on different tasks.** The horizontal dashed line indicates the best performance over all methods. DV (Diffusion planning) stands out in Kitchen, Maze2D, and AntMaze; while DQL (Diffusion policy) (Wang et al., 2023b) outperforms all diffusion planning methods in MuJoCo locomotion tasks. Refer to the caption of Table 1 for method details.

Diffusion planning and diffusion policy represent two key approaches within diffusion-based decision-making. After examining the core components of diffusion planners, we turn to a comparison of diffusion planning and diffusion policy across different environments. The experimental results are illustrated in Fig. 8. We observed that diffusion planning outperforms diffusion policy in AntMaze, Kitchen, and Maze2D, whereas diffusion policy excels in MuJoCo locomotion tasks. The first three environments require precise goal achievement, such as positioning an object exactly, necessitating long-term planning. This makes them well-suited to diffusion planning, which generates entire trajectories in one step. Furthermore, these environments feature sparse reward structures, posing challenges for model-free RL algorithms typically used in diffusion policies (Wang et al., 2023b). In contrast, the objective in MuJoCo is simply to control agents to run faster, a task that is less related to lookahead planning and does not require intricate planning. RL loss functions can help diffusion policy (Wang et al., 2023b) achieve better results in such scenarios.

## 4.7 VALIDATIONS ON ADROIT DATASET

To examine whether the conclusions drawn from our experiments can generalize to other tasks, we conducted experiments on the Adroit Hand dataset (Rajeswaran et al., 2018; Fu et al., 2020), which features motion-captured human data applied to a realistic, high-degree-of-freedom robotic hand, including both challenges from planning and control. It encompasses 8 challenging tasks highlighted in the original paper, including as pen twirling, door opening, hammer use, and object relocation. We found that the results are consistent with our findings, supporting the generalizablity across tasks. The detailed results are deferred to Appendix C.

## 4.8 PRACTICAL TIPS TO TAKE HOME

**Takeaway 1:** Diffusion planning is most effective for tasks requiring long-term credit assignment, while diffusion policies better fit locomotion tasks that demand less long-term planning (Sect. 4.6)

**Takeaway 2:** It is recommended to generate state plans with diffusion planners and use an inverse dynamics model to compute the corresponding actions (Sect. 4.1).

**Takeaway 3:** Implementing jump-step planning can be highly beneficial; experimenting with different planning strides is encouraged (Sect. 4.2).

**Takeaway 4:** It is worth trying to use Transformer as the backbone of diffusion planner, especially in

the tasks that require long-term lookahead planning (Sect. 4.3).

**Takeaway 5:** A single-layer Transformer is insufficient for effective planning (Sect. 4.4).

**Takeaway 6:** Larger models do not necessarily lead to better performance in diffusion planner for offline RL (Sect. 4.4).

**Takeaway 7:** Non-guidance approaches, such as Monte Carlo unconditional sampling with selection, can outperform classifier or classifier-free guidance when the dataset contains enough near-optimal trajectories (Sect. 4.5).

## 5 DISCUSSIONS

**Synergy between diffusion planning and diffusion policy.** A significant avenue for future research involves a deeper exploration of the distinctions between diffusion planning and diffusion policy. Drawing on Daniel Kahneman's seminal work *Thinking, Fast, and Slow* (Kahneman, 2011), human cognitive processes are categorized into System 1 and System 2. Diffusion policies are analogous to System 1 processes, as they operate rapidly and efficiently (Wang et al., 2023b), making them well-suited for tasks such as locomotion (Fig. 8) that do not require extensive deliberation or long-term planning. These policies manage routine decision making with the same efficiency as intuitive responses in human cognition. Conversely, diffusion planning mirrors System 2 thinking, characterized by its slower, more deliberate, and effortful nature. This approach is particularly effective for tasks that demand long-term credit assignment (Fig. 8), involving more computations to develop effective plans. In RL terminology, diffusion planning can be broadly classified as model-based, while diffusion policy aligns with model-free methodologies. Investigating the interplay between these two systems presents a compelling intersection for both machine learning and cognitive neuroscience (Gläscher et al., 2010; Duan et al., 2016; Botvinick et al., 2019). Studies from cognitive science indicate that the brain may use a synergistic approach which arbitrates and selects the better system according to the current situation, and the preference may change over time (Lee et al., 2014; Han et al., 2024). We anticipate extensive future research focused on integrating the strengths of diffusion planning and diffusion policies to enable both efficient and effective decision-making AI.

**Computational efficiency.** Despite the effectiveness of diffusion planners, their computational cost is substantial. Our work is orthogonal to the optimization of computational cost (Dong et al., 2024a). Nonetheless, future work may consider new schemes such as the consistency model (Song et al., 2023) to improve computational efficiency.

**Interpretability and safety.** Our study focuses on a single performance metric (total return), potentially overlooking qualitative aspects such as the interpretability and reliability of the diffusion planner. Future work may consider issues such as explainability (Puiutta & Veith, 2020) and safety (Xiao et al., 2023) of diffusion planning. Leveraging the experiences from computer vision domain will be worth investigating.

**Sustainability.** Our work required significant computational resources, particularly in terms of GPU energy consumption, as we trained and evaluated thousands of models across diverse tasks. However, this investment in energy is not without purpose. We aim to provide a solid foundation for future research. Subsequent work can build upon our findings, reducing the need for extensive trial-and-error experimentation. In this way, our research contributes to energy efficiency in the long term, as researchers can reference our results and apply proven methods rather than duplicating resource-intensive exploratory efforts.

**Open problems and future directions.** In the current study, we have focused on standard Markov decision process problems (Bellman, 1957) using a popular offline RL benchmark (Fu et al., 2020). The planning and control are based on joint states and coordinates. Numerous untouched problems exist, such as vision-based decision making (Du et al., 2024; Yang et al., 2024), goal-conditioned reinforcement learning (Liu et al., 2022; Wang et al., 2023a), partially observable environments (Schmidhuber, 1991), offline-to-online deployment (Matsushima et al., 2021), and the scalability of diffusion planning models (Kaplan et al., 2020). Future efforts are anticipated to fully address these limitations. However, even within the scope of the current work, we have found several interesting phenomena and tips that are counter to common practices. Our work should be considered as a new but solid starting point for behavior planning using decision models.

## REPRODUCIBILITY STATEMENT

We are committed to ensuring the reproducibility of our results. To facilitate this, we include the source code of DV in the supplementary material, which is also avaliable at `https://github.com/Josh00-Lu/DiffusionVeteran`. Detailed descriptions of our experimental setup, including model architectures, training procedures, and hyperparameter settings, are provided in Appendix A and Appendix B. We have included comprehensive information on the datasets used, along with any preprocessing steps, in Appendix B. For all key experiments, we have specified the evaluation protocols and metrics in Sect. 3 and provided extensive results in Appendix D. We have included the full list of hyperparameters and configurations in Appendix B.4.

## ACKNOWLEDGMENT

This work is supported by Microsoft Research.

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

# A  GUIDED SAMPLING ALGORITHMS

For decision making tasks, guided sampling algorithms are used to generate desired plans or actions. In this work, we compare three types of different guided sampling methods: classifier guidance (Dhari-wal & Nichol, 2021) (CG), classifier-free guidance (CFG) (Ho & Salimans, 2021), and Monte Carlo sampling from selections (MCSS).

**Classifier guidance:**  Classifier guidance (CG) is introduced to guide the unconditional diffusion models $q_t(\boldsymbol{x}_t)$ to generate data over condition $\boldsymbol{c}$. The conditioned score function is formulated as:

$$\nabla_{\boldsymbol{x}} \log q_t(\boldsymbol{x}_t|\boldsymbol{c}) = \nabla_{\boldsymbol{x}} \log q_t(\boldsymbol{x}_t) + \nabla_{\boldsymbol{x}} \log q_t(\boldsymbol{c}|\boldsymbol{x}_t)$$

where the second term is also know as a noised classifier that predict the condition using noised data $\boldsymbol{x}_t$. During sampling, the gradient of the classifier is then applied to the predicted noise $\epsilon_\theta(\boldsymbol{x}_t, t)$:

$$\bar{\epsilon}_\theta(\boldsymbol{x}_t, t, \boldsymbol{c}) = \epsilon_\theta(\boldsymbol{x}_t, t) - w\sigma_t \nabla_{\boldsymbol{x}} \log q_t(\boldsymbol{c}|\boldsymbol{x}_t)$$

where $w$ is a weighting factor that controls the strength of the classifier guidance. For CG sampling, we tuned $w$ in range $[0.001, 10]$ on each task.

**Classifier-free guidance:**  To avoid training classifiers, classifier-free guidance (CFG) is proposed. The main idea of CFG is to train a diffusion model that can be used for both conditional noise predictor $\epsilon_\theta(\boldsymbol{x}_t, t, \boldsymbol{c})$ and unconditional noise predictor $\epsilon_\theta(\boldsymbol{x}_t, t)$:

$$\bar{\epsilon}_\theta(\boldsymbol{x}_t, t, \boldsymbol{c}) = \epsilon_\theta(\boldsymbol{x}_t, t) + w(\epsilon_\theta(\boldsymbol{x}_t, t, \boldsymbol{c}) - \epsilon_\theta(\boldsymbol{x}_t, t))$$

where $\epsilon_\theta(\boldsymbol{x}_t, t) = \epsilon_\theta(\boldsymbol{x}_t, t, \varnothing)$. Noise prediction of $\epsilon_\theta(\boldsymbol{x}_t, t, \varnothing)$ and $\epsilon_\theta(\boldsymbol{x}_t, t, \boldsymbol{c})$ can be jointly learned by randomly discard conditioning with probability of $p_{\text{uncond}}$. For decision making tasks, we can train diffusion models using condition of discounted returns, and using classifier-free guidance for better plan sampling. We can normalize the discounted return in the dataset for training, and use condition of 1 as target return for CFG sampling during inference (Ajay et al., 2022). However, experiments shows that fixing 1 as target may lead to unrealistic or unstable plans. Consequently, besides tuning the guidance strength $w \in [1.0, 6.0]$, we also tune for the best target return within range $[0.5, 1.5]$ for each tasks to test CFG's best performance.

**Monte Carlo sampling from selections:**  For Monte Carlo sampling from selections (MCSS), $N$ selections are firstly sampled from an unconditional generative model as candidates. Then these candidates are evaluated with a learned critic for the selection of the optimal one. One advantage for MCSS is that, it do not rely on any task-specific hyperparameters for inference, such as the guidance strength $w$ in CFG and CG, and target return in CFG. However, it needs to sample $N - 1$ more candidates during each decision making step.

# B  IMPLEMENTATION DETAILS

## B.1  MODEL ARCHITECTURE

**Planner:**  Our code is based on CleanDiffuser (Dong et al., 2024b). We examined U-Net (Ron-neberger et al., 2015) and Transformer (Vaswani et al., 2017) as the neural network backbone for all the diffusion planners. Specifically, we keep consistent with the implementation of U-Net1D (Janner et al., 2022), with 5 kernel size, $(1, 2, 2, 2)$ for channel multiplication, 32 base channels on MuJoCo, Kitchen and Maze2D, and 64 base channels on AntMaze. For Transformer, we use DiT1D (Peebles & Xie, 2023; Dong et al., 2024c), with hidden dimension of 256, head dimension of 32, 2 DiT blocks on MuJoCo, Kitchen and Maze2D, and 8 DiT blocks on AntMaze. All the planner diffusion models are trained with the Adam (Kingma & Ba, 2014) optimizer with learning rate of $3e - 4$, batch size of 128, for 1M gradient steps. All the diffusion models in this work are trained to predict the noise. However, for U-Net1D experiments on Kitchen, the diffusion planner to predict the clean estimation, because it could achieve quite better performance.

**Inverse dynamics:**  We used an MLP-based diffusion model as the inverse dynamic model, whose input is the current state and the planned next-state; and the output is the action to execute. It is implemented with a 3-layer MLP with additional 2-layer embedding layer and trained with $1M$ gradient steps. All the inverse dynamic models are trained with the Adam (Kingma & Ba, 2014) optimizer with learning rate of $3e - 4$, batch size of 128. We found that the using diffusion model

as inverse dynamics show similar performance with vanilla MLP inverse dynamics. The inverse dynamics trained on navigation tasks (Maze2D and AntMaze) are trained with policy centralization, where the original $(s_t, s_{t+1}) \rightarrow a_t$ is modified with $(0, s_{t+1} - s_t) \rightarrow a_t$. We found this may improve the generalization ability of the inverse dynamics on navigation tasks.

**Critic:** We also implemented two types of critic models for guided sampling. The first type has the architecture of the U-Net1D for the planner, with a linear output layer to produce the critic value. The second type is a 2 blocks vanilla transformer, with hidden dimension of 256 as the value function, with a linear projection head on the first token output. We trained all the critic model using the Adam (Kingma & Ba, 2014) optimizer with learning rate of $3e - 4$, batch size of 128. The model will be trained with 200K gradient steps, if it is a clean critic model [1]. Otherwise, it is trained for 1M gradient steps.

## B.2 DIFFUSION SOLVER

We use DDIM (Song et al., 2020) of temperature 1.0 for planner diffusion sampling, and DDPM with temperature of 0.5 for inverse dynamic action sampling. The sampling temperature is introduced to reduce sampling randomness (Ajay et al., 2022).

## B.3 DATASET PRE-PROCESSING

Diffusion policy baselines (Chen et al., 2023; Wang et al., 2023b; Hansen-Estruch et al., 2023) commonly learns policy, Q functions, and value functions using a temporal difference manner, on standard transitions $(s_t, a_t, s_{t+1}, r_t)$. Admittedly, diffusion planners often require careful dataset pre-processing, including horizon padding, planning strides, return calculation, and truncation-termination handling. An unsuccessful sequential dataset pre-processing may greatly reduce the planning ability of diffusion planners. Most planning tasks are usually sparse rewarded, and how optimality is defined, combined with different temporal credit assignment methods is also important. For MuJoCo and Kitchen, we use discount factor $\gamma = 0.997$. For Maze2D and AntMaze, we use IQL-maze (Kostrikov et al., 2021) reward shaping methods for temporal credit assignment in navigation planning tasks, where an $-1$ penalty is always applied to agent during every timestep. The plan trajectory is clipped to 1000 steps on Maze2D. Refer to our code for more details.

## B.4 FULL HYPER-PARAMETERS

We conducted several rounds of hyperparameter tuning, where each round conducted grid search on a subset of hyperparameters that we identified as most influential based on prior experiments and domain knowledge. We control this iterative process of selecting which hyperparameters to explore in each round, guided by preliminary results and insights. Table 2 displays the hyperparameters and default choices in our work.

---

[1] A clean critic model is only trained on the original input data without noise, while a noised critic model is trained using noised data using the diffusion model's noise schedule

Table 2: Configuration Settings

| Settings | Default | Choices |
|---|---|---|
| Guidance Type | MCSS | [MCSS, CG, CFG, None] |
| State-Action Generation | Separate | [Joint, Separate] |
| Advantage Weighting | True only on MuJoCo | [True, False] |
| Inverse Dynamic | Diffusion | [Diffusion, Regular] |
| Time Credit Assignment | discount=0.997 | [discount=0.997, IQL-maze] |
| Planner Net. Backbone | Transformer | [Transformer, UNet] |
| UNet Channels Mult | (1, 2, 2, 2) | (1, 2, 2, 2) |
| UNet Base Channels | 32 | [16, 32, 64] |
| Transformer Hidden | 256 | 256 |
| Transformer Block | 2 | [2, 4, 6, 8] |
| Planner Solver | DDIM | [DDIM, DDPM] |
| Planner Sampling Steps | 20 | 20 |
| Planner Training Steps | 1000000 | 1000000 |
| Planner Temperature | 1 | 1 |
| MCSS Candidates | 50 | [1, 20, 50] |
| Planning Horizon | 32 | [4, 32, 40] |
| Planning Stride | 1 | [1, 2, 4, 5, 15, 25] |
| Inverse Dynamics Net. Backbone | MLP | MLP |
| Inverse Dynamics Hidden | 256 | 256 |
| Inverse Dynamics Solver | DDPM | DDPM |
| Inverse Dynamics Sampling Steps | 10 | 10 |
| Inverse Dynamics Training Steps | 1000000 | 1000000 |
| Policy Temperature | 0.5 | 0.5 |

## C    RESULTS ON VALIDATION DATASET

In this section, we validate our insights and findings on a new set of eight tasks, called Adroit Hand (Rajeswaran et al., 2018; Fu et al., 2020), to test the generalizability of our conclusions regarding diffusion planning derived from the main paper.

### C.1    EXPERIMENT SETUPS

As demonstrated in Fig. 9, there are four different types of challenging tasks in the Adroit Hand environments. Each task consists of a dexterous hand attached to a free arm, which has around 30 actuated degrees of freedom for controlling and moving to complete different manipulation tasks, including opening the door, driving the nail, repositioning the pen orientation, and relocating the ball.

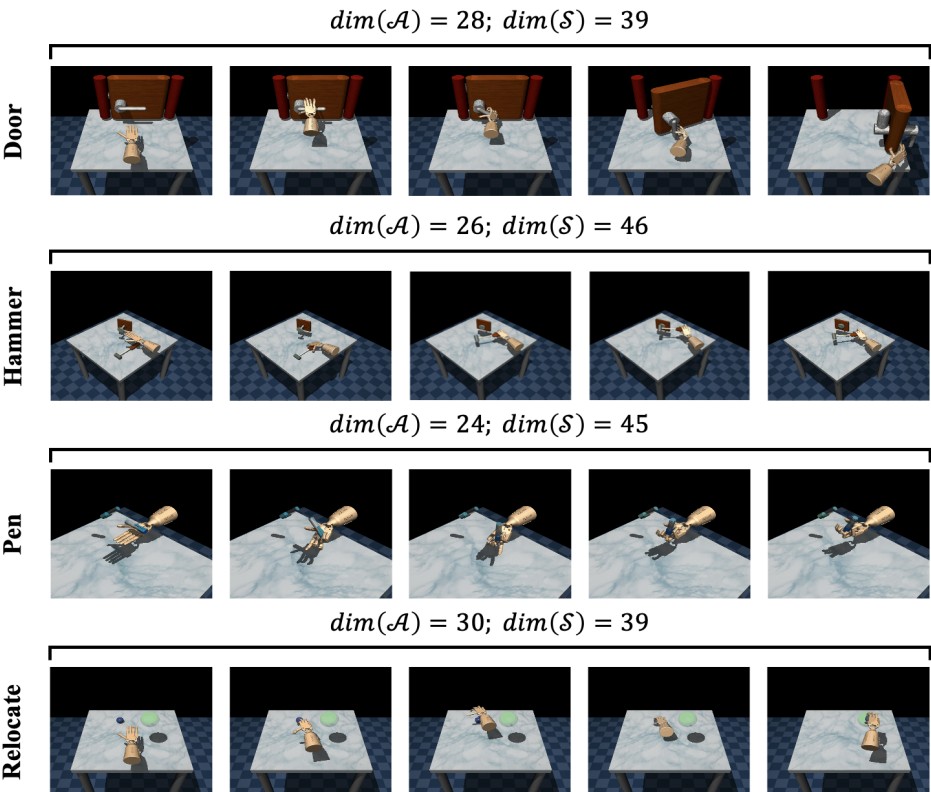

Figure 9: Rendering of the validation benchmarking tasks of Adroit Hand, where $dim(\mathcal{S})$ and $dim(\mathcal{A})$ denote the dimension of the state and action spaces on each tasks.

**Door**  The task consists of undoing the latch and swinging the door open. The latch has significant dry friction and a bias torque that forces the door to remain closed. The agent leverages environmental interaction to develop an understanding of the latch, as no information about the latch is explicitly provided. The position of the door is randomized. The task is considered complete when the door touches the door stopper at the other end.

**Hammer**  The task involves picking up a hammer and driving a nail into a board. The nail position is randomized and has dry friction capable of absorbing up to 15N of force. The task is successful when the entire length of the nail is inside the board.

**Pen**  The task requires repositioning the blue pen to match the orientation of the green target. The base of the hand is fixed, and the target is randomized to cover all configurations. The task is considered successful when the orientations match within a specified tolerance.

**Relocate** The task involves moving the blue ball to the green target. The positions of the ball and target are randomized over the entire workspace. The task is considered successful when the object is within an epsilon-ball of the target.

We conduct our experiments on two types of datasets: Cloned and Expert. The Cloned dataset consists of a 50-50 split between demonstration data and 2,500 trajectories sampled from a behaviorally cloned policy trained on these demonstrations. The demonstration data includes 25 human trajectories, which are duplicated 100 times to match the number of cloned trajectories. The Expert dataset comprises 5,000 trajectories sampled from an expert policy that successfully solves the task, as provided in the DAPG repository.

## C.2 BASELINES

In order to better validate the performance of our diffusion veteran (DV), we also re-implement four representative baselines on this new set of environments: (i) *Diffusion policies*: DQL (Wang et al., 2023b) and IDQL (Hansen-Estruch et al., 2023); (ii) *Diffusion planners*: DD (Ajay et al., 2022) and Diffuser (Janner et al., 2022).

## C.3 EXPERIMENTAL RESULTS

In this section, we validate all our insights and findings of diffusion planning in the main paper from the same five perspectives: (1) **Action Generation**, (2) **Planning Strategy**, (3) **Impact of Network Size**, (4) **Denoising Network Backbone**, and (5) **Guided Sampling Algorithms** to better assess the generalizability of the main paper.

### C.3.1 ACTION GENERATION

Table 3: Results of different action generation choices for diffusion planning on Adroit Hand. Data are Mean $\pm$ Standard Error over 150 episode seeds.

| Environment | Dataset | Separate (DV) | Joint |
|:---:|:---:|:---:|:---:|
| door | cloned | 1.5 ± 0.0 | 15.2 ± 0.4 |
| door | expert | 104.7 ± 0.5 | 104.7 ± 0.2 |
| hammer | cloned | 11.9 ± 0.7 | 2.6 ± 0.0 |
| hammer | expert | 125.8 ± 1.1 | 113.4 ± 1.1 |
| pen | cloned | 80.2 ± 2.0 | 85.2 ± 2.0 |
| pen | expert | 122.2 ± 1.8 | 112.7 ± 1.8 |
| relocate | cloned | 0.6 ± 0.0 | 0.4 ± 0.0 |
| relocate | expert | 108.9 ± 0.2 | 108.7 ± 0.3 |
| **Average** | | **69.5** | 67.9 |

The experimental results presented in Table 3 corroborate our previous findings regarding action generation strategies in diffusion planning. Specifically, generating state plans using diffusion planners and then computing the corresponding actions via an inverse dynamics model (the Separate (DV) approach) demonstrates superior or comparable performance to the Joint approach, which involves generating actions directly. Averaged across all tasks and datasets, the Separate (DV) method achieves a higher mean score of 69.5 compared to 67.9 for the Joint method. This overall performance gain underscores the effectiveness of decoupling state planning from action generation, allowing the diffusion model to focus on modeling state distributions more accurately.

### C.3.2 PLANNING STRATEGY

Table 4: Results of different planning strategy choices on Adroit Hand. Data are Mean ± Standard Error over 150 episode seeds.

| Environment | Dataset | Stride 1 | Stride 2 (DV) | Stride 4 |
|---|---|---|---|---|
| door | cloned | 13.6 ± 0.4 | 1.5 ± 0.0 | 0.1 ± 0.0 |
| door | expert | 104.6 ± 0.2 | 104.7 ± 0.5 | 105.6 ± 0.2 |
| hammer | cloned | 3.9 ± 0.2 | 11.9 ± 0.7 | 3.2 ± 0.8 |
| hammer | expert | 112.5 ± 1.2 | 125.8 ± 1.1 | 125.9 ± 1.5 |
| pen | cloned | 81.6 ± 2.0 | 80.2 ± 2.0 | - - |
| pen | expert | 125.9 ± 1.6 | 122.2 ± 1.8 | - - |
| relocate | cloned | 0.1 ± 0.0 | 0.6 ± 0.0 | 0.0 ± 0.0 |
| relocate | expert | 108.0 ± 0.3 | 108.9 ± 0.2 | 109.0 ± 0.6 |
| **Average** | | 68.8 | **69.5** | - - |

The results in Table 4 show that implementing jump-step planning strategies can enhance the performance of diffusion planning. On average, planning with a stride of 2 achieves a higher mean score compared to stride 1, indicating the benefits of experimenting with different strides. Notably, the "pen" environment has a maximum of 100 steps, which does not support planning with strides greater than 4; however, even within these constraints, stride planning demonstrates performance improvements. These findings suggest that increasing the planning stride can be beneficial for diffusion planning.

### C.3.3 DENOISING NETWORK BACKBONE

Table 5: Results of different denoising network backbone choices on Adroit Hand. Data are Mean ± Standard Error over 150 episode seeds.

| #Model Parameters | | 2.64 M | 3.96 M | 15.80 M | 63.11 M |
|---|---|---|---|---|---|
| **Environment** | **Dataset** | **Transformer (DV)** | UNet | UNet | UNet |
| door | cloned | 1.5 ± 0.0 | -0.2 ± 0.0 | -0.2 ± 0.0 | 1.2 ± 0.4 |
| door | expert | 104.7 ± 0.5 | -0.1 ± 0.0 | -0.0 ± 0.0 | 103.7 ± 0.6 |
| hammer | cloned | 11.9 ± 0.7 | -0.2 ± 0.0 | -0.0 ± 0.0 | 1.6 ± 0.0 |
| hammer | expert | 125.8 ± 1.1 | -0.1 ± 0.0 | -0.0 ± 0.0 | 122.0 ± 1.8 |
| pen | cloned | 80.2 ± 2.0 | -0.7 ± 0.2 | -1.8 ± 0.3 | 73.4 ± 5.1 |
| pen | expert | 122.2 ± 1.8 | -1.3 ± 0.1 | -2.6 ± 0.2 | 134.0 ± 3.2 |
| relocate | cloned | 0.6 ± 0.0 | -0.1 ± 0.0 | -0.1 ± 0.0 | 0.0 ± 0.0 |
| relocate | expert | 108.9 ± 0.2 | -0.1 ± 0.0 | -0.1 ± 0.0 | 106.5 ± 0.9 |
| **Average** | | **69.5** | -0.4 | -0.6 | 67.8 |

The results in Table 5 show that when using a regular number of parameters, Transformers have a clear advantage over UNet as the denoising backbone in diffusion planning. Specifically, the Transformer model achieves an average score of 69.5 with only 2.64 M parameters, while UNet needs significantly more parameters (up to 63.11 M, around **25 times** of the transformer) to reach a similar performance level (average score of 67.8). This demonstrates that UNet requires multiple times the parameters to match the efficiency of Transformers.

### C.3.4 IMPACT OF NETWORK SIZE

Table 6: Performance change over depth of the Transformer network for diffusion planner on Adroit Hand. Data are Mean ± Standard Error over 150 episode seeds.

| #Model Parameters | | 1.46 M | 2.64 M | 3.82 M |
|---|---|---|---|---|
| **Environment** | **Dataset** | Depth 1 | **Depth 2 (DV)** | Depth 3 |
| door | cloned | 4.3 ± 0.9 | 1.5 ± 0.0 | 1.3 ± 0.1 |
| door | expert | 0.0 ± 0.0 | 104.7 ± 0.5 | 105.5 ± 0.4 |
| hammer | cloned | 17.0 ± 2.4 | 11.9 ± 0.7 | 1.0 ± 0.0 |
| hammer | expert | 76.1 ± 4.9 | 125.8 ± 1.1 | 126.0 ± 1.4 |
| pen | cloned | 44.9 ± 5.3 | 80.2 ± 2.0 | 75.7 ± 5.4 |
| pen | expert | 42.5 ± 4.9 | 122.2 ± 1.8 | 127.5 ± 4.1 |
| relocate | cloned | 0.7 ± 0.1 | 0.6 ± 0.0 | 0.0 ± 0.0 |
| relocate | expert | 0.8 ± 0.4 | 108.9 ± 0.2 | 107.4 ± 0.8 |
| **Average** | | 23.3 | **69.5** | 68.1 |

The results presented in Table 6 demonstrate that a single-layer Transformer (Depth 1) is inadequate for effective planning, as evidenced by its significantly lower average score of 23.3 compared to deeper models. When the depth is increased to two layers (Depth 2), the performance improves markedly, achieving an average score of 69.5. However, further increasing the depth to three layers (Depth 3) does not yield additional benefits; the average score slightly decreases to 68.1. These findings support our earlier observations: a single-layer Transformer is insufficient for planning tasks, and simply enlarging the model does not guarantee better performance in diffusion planning for offline reinforcement learning.

### C.3.5 GUIDED SAMPLING ALGORITHMS

The results in Table 7 indicate that non-guidance methods like Monte Carlo sampling with selection (MCSS) can outperform guidance-based approaches when the dataset contains sufficient near-optimal trajectories. MCSS achieves the highest average score of 69.5, surpassing classifier-free guidance (CFG) at 67.7, classifier guidance (CG) at 62.0, and unguided sampling (None) at 60.9.

Table 7: Results of different guided sampling algorithms on Adroit Hand. Data are Mean ± Standard Error over 150 episode seeds.

| **Environment** | **Dataset** | **MCSS (DV)** | CFG | CG | None |
|---|---|---|---|---|---|
| door | cloned | 1.5 ± 0.0 | 12.1 ± 0.1 | 0.9 ± 0.0 | 0.7 ± 0.1 |
| door | expert | 104.7 ± 0.5 | 103.5 ± 0.4 | 104.3 ± 0.1 | 103.7 ± 0.2 |
| hammer | cloned | 11.9 ± 0.7 | 7.6 ± 0.1 | 1.2 ± 0.0 | 0.4 ± 0.0 |
| hammer | expert | 125.8 ± 1.1 | 106.7 ± 0.9 | 110.4 ± 1.3 | 105.7 ± 1.3 |
| pen | cloned | 80.2 ± 2.0 | 74.7 ± 1.6 | 64.3 ± 1.9 | 64.1 ± 2.0 |
| pen | expert | 122.2 ± 1.8 | 128.2 ± 1.4 | 107.9 ± 1.9 | 105.8 ± 1.8 |
| relocate | cloned | 0.6 ± 0.0 | 0.7 ± 0.0 | 0.0 ± 0.0 | -0.0 ± 0.0 |
| relocate | expert | 108.9 ± 0.2 | 108.2 ± 0.5 | 107.0 ± 0.3 | 106.8 ± 0.3 |
| **Average** | | **69.5** | 67.7 | 62.0 | 60.9 |

### C.3.6 COMPARISON WITH OTHER METHODS

Table 8: Performance comparison with representative baselines on Adroit Hand. Data are Mean ± Standard Error over 150 episode seeds.

| Tasks | | Diffusion Policies | | | Diffusion Planners | | |
|---|---|---|---|---|---|---|---|
| Dataset | Environment | IDQL | DQL-TuneLR | DQL | Diffuser | DD | DV (Ours) |
| door | cloned | 4.4 ± 0.6 | -0.3 ± 0.0 | -0.1 ± 0.0 | 0.1 ± 0.1 | 15.4 ± 0.5 | 1.5 ± 0.0 |
| door | expert | 105.0 ± 0.3 | 104.8 ± 0.3 | 104.3 ± 0.1 | 103.0 ± 0.5 | 105.5 ± 0.3 | 104.7 ± 0.5 |
| hammer | cloned | 3.5 ± 0.5 | 0.2 ± 0.0 | 0.1 ± 0.0 | 1.2 ± 0.1 | 1.6 ± 0.1 | 11.9 ± 0.7 |
| hammer | expert | 127.6 ± 0.1 | 128.3 ± 0.1 | 55.9 ± 5.2 | 103.1 ± 3.8 | 124.8 ± 2.1 | 125.8 ± 1.1 |
| pen | cloned | 82.3 ± 5.0 | 23.3 ± 4.0 | 28.3 ± 4.3 | 61.7 ± 5.0 | 72.0 ± 4.2 | 80.2 ± 2.0 |
| pen | expert | 137.8 ± 2.4 | 133.5 ± 3.9 | 60.9 ± 6.1 | 99.7 ± 4.8 | 139.8 ± 3.5 | 122.2 ± 1.8 |
| relocate | cloned | 0.0 ± 0.1 | 0.1 ± 0.0 | -0.1 ± 0.0 | -0.0 ± 0.0 | 0.3 ± 0.0 | 0.6 ± 0.0 |
| relocate | expert | 107.0 ± 0.8 | 108.5 ± 0.6 | 108.8 ± 0.6 | 102.2 ± 1.5 | 110.3 ± 1.1 | 108.9 ± 0.2 |
| **Average** | | 71.0 | 62.3 | 44.8 | 58.9 | **71.2** | 69.5 |

Finally, we conduct a comprehensive comparison with our re-implemented baselines (Table 8). Notably, we find that using the default learning rate for DQL in this environment may lead to performance degradation. Therefore, we perform a learning rate search for DQL, select the optimal one, and denote it as DQL-TuneLR, where `learning_rate` $= \{3e-3, 3e-4, 3e-5\}$. For IDQL, we use the officially recommended 256 candidates for high-density action-value estimation. For DD, we conduct a grid search over 35 possible configurations for each task, adjusting `target_return` $= \{0.4, 0.6, 0.8, 1.0, 1.2\}$ and `w_cfg` $= \{1.0, 1.5, 2.0, 2.5, 3.0, 3.5, 4.0\}$ to guarantee its maximum performance. Our method requires only one-quarter of the candidates and no task-specific fine-tuning to achieve comparable performance.

# D EXTENSIVE RESULTS

## D.1 ATTENTION MAP ACROSS DIFFERENT TASKS

The following plots show some examples of the attention weights in different DDIM denoising steps in each task using Transformer backbones. We can see that a long-term dependency is generally existing in the Transformer.

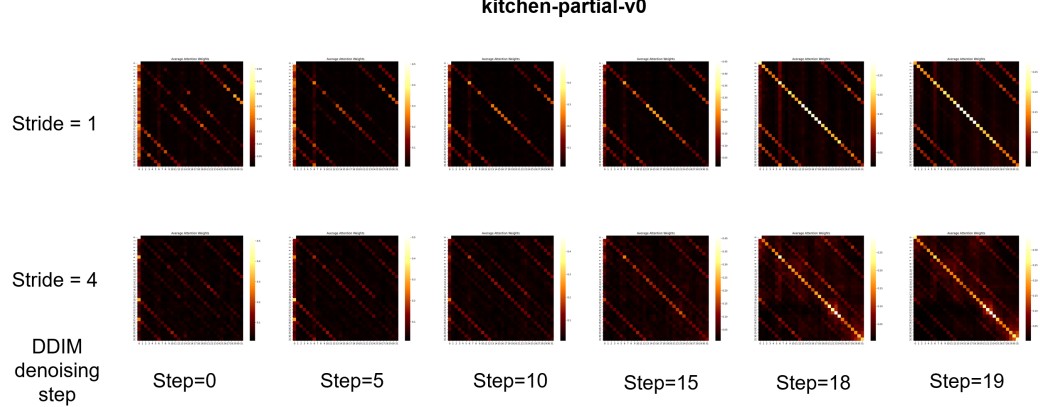

Figure 10: Attention weights (averaged on multi-heads) of the first Transformer layer of DV on the Kitchen-Partial-v0 dataset.

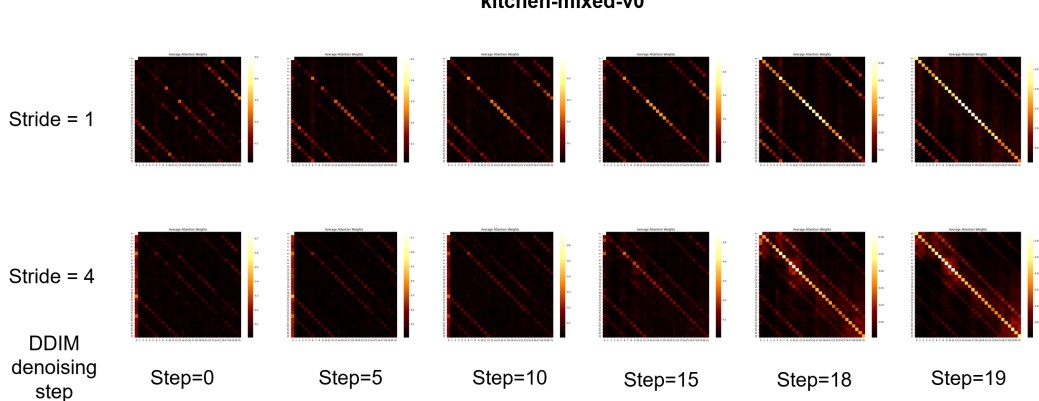

Figure 11: Attention weights (averaged on multi-heads) of the first Transformer layer of DV on the Kitchen-Mixed-v0 dataset.

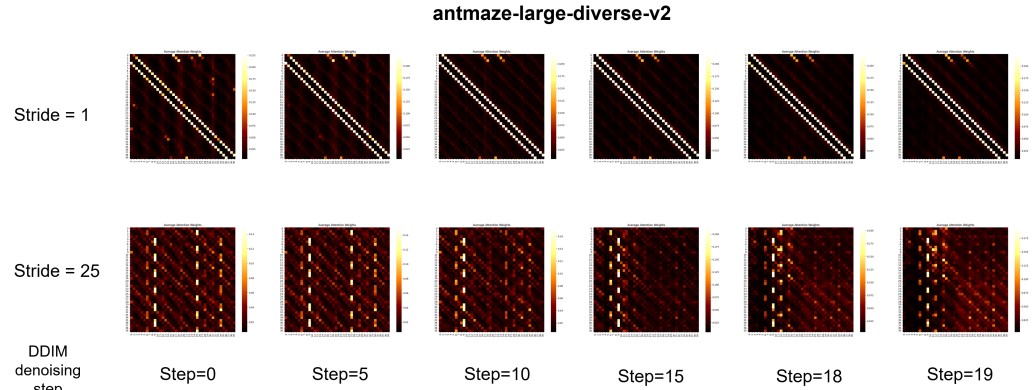

Figure 12: Attention weights (averaged on multi-heads) of the first Transformer layer of DV on the AntMaze-Large-Diverse-v2 dataset.

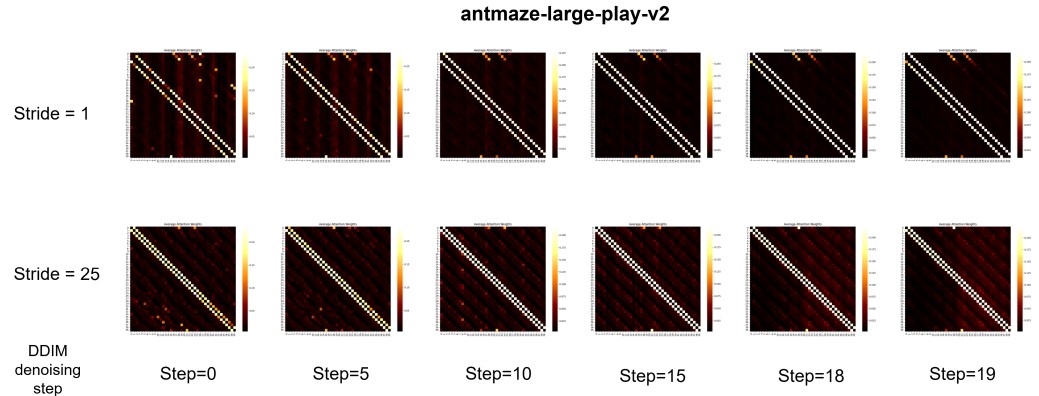

Figure 13: Attention weights (averaged on multi-heads) of the first Transformer layer of DV on the AntMaze-Large-Play-v2 dataset.

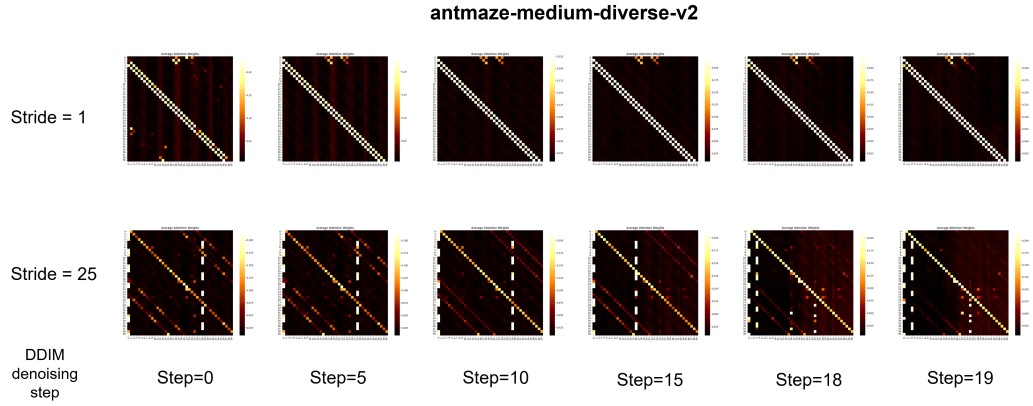

Figure 14: Attention weights (averaged on multi-heads) of the first Transformer layer of DV on the AntMaze-Medium-Diverse-v2 dataset.

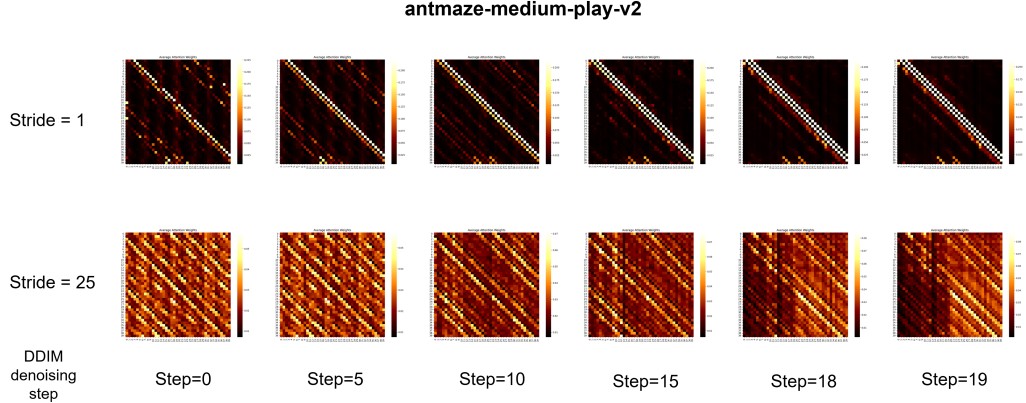

Figure 15: Attention weights (averaged on multi-heads) of the first Transformer layer of DV on the AntMaze-Medium-Play-v2 dataset.

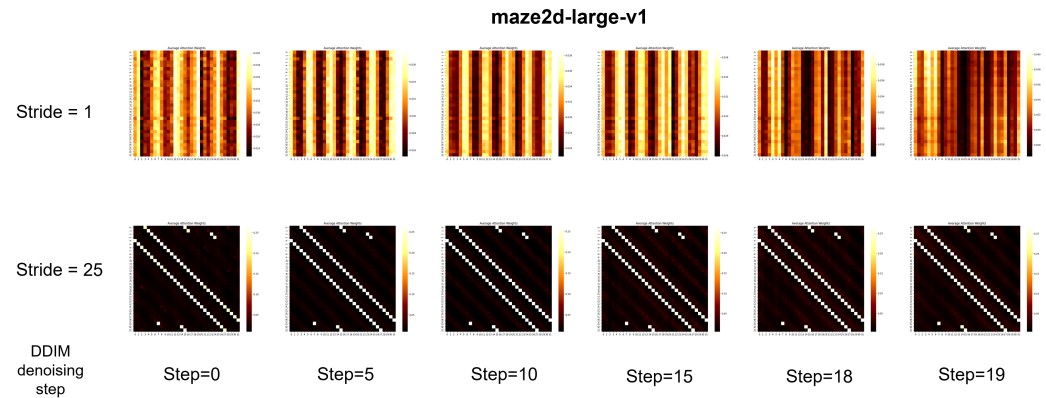

Figure 16: Attention weights (averaged on multi-heads) of the first Transformer layer of DV on the Maze2d-Large-v1 dataset.

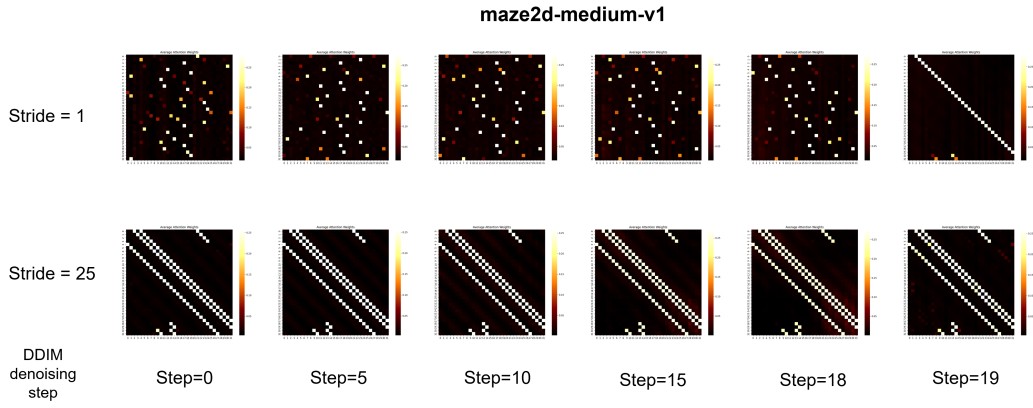

Figure 17: Attention weights (averaged on multi-heads) of the first Transformer layer of DV on the Maze2D-Medium-v1 dataset.

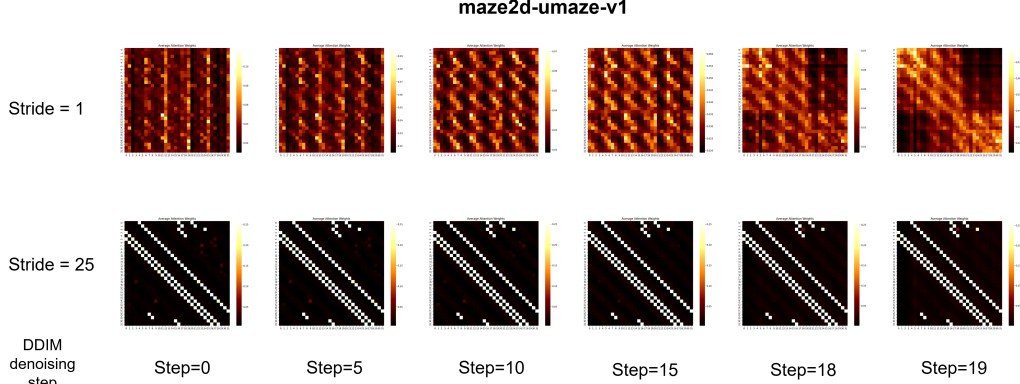

Figure 18: Attention weights (averaged on multi-heads) of the first Transformer layer of DV on the Maze2D-Umaze-v1 dataset.

## D.2 Extensive Results

Table 9 (part 1): Guided sampling algorithms **MCSS** and **CFG**.

| Dataset | Environment | MCSS Stride 1 | MCSS Stride 2/5 | MCSS Stride 4/15 | MCSS Stride 8/25 | CFG Stride 1 | CFG Stride 2/5 | CFG Stride 4/15 | CFG Stride 8/25 |
|---|---|---|---|---|---|---|---|---|---|
| Mixed | Kitchen | 67.2 ± 0.3 | 74.2 ± 0.1 | 73.6 ± 0.1 | — | 72.7 ± 0.2 | 78.6 ± 0.1 | 72.4 ± 0.1 | — |
| Partial | Kitchen | 87.8 ± 0.6 | 88.3 ± 0.4 | 94.0 ± 0.3 | — | 95.9 ± 0.4 | 98.6 ± 0.3 | 95.2 ± 0.6 | — |
| **Average** | | 77.5 | 81.3 | 83.8 | — | 84.3 | 88.6 | 83.8 | — |
| Antmaze-Large | Diverse | 72.4 ± 1.4 | 2.6 ± 0.1 | 71.2 ± 2.0 | 80.0 ± 1.8 | 54.4 ± 0.6 | 46.5 ± 0.1 | 6.4 ± 0.1 | 67.4 ± 0.3 |
| Antmaze-Large | Play | 61.2 ± 1.5 | 18.2 ± 0.5 | 76.2 ± 0.9 | 76.4 ± 2.0 | 60.8 ± 0.8 | 0.4 ± 0.0 | 59.4 ± 0.3 | 56.4 ± 0.5 |
| Antmaze-Medium | Diverse | 58.8 ± 1.5 | 24.2 ± 0.8 | 86.2 ± 1.6 | 87.4 ± 1.6 | 57.1 ± 0.6 | 5.0 ± 0.1 | 33.0 ± 0.8 | 49.0 ± 0.7 |
| Antmaze-Medium | Play | 38.8 ± 1.5 | 73.2 ± 1.5 | 83.8 ± 1.9 | 89.0 ± 1.6 | 55.9 ± 0.8 | 51.3 ± 0.7 | 28.7 ± 0.4 | 29.4 ± 0.5 |
| **Average** | | 57.8 | 29.6 | 79.4 | 83.2 | 57.1 | 25.8 | 31.9 | 50.6 |
| Maze2D | Umaze | 131.8 ± 1.3 | 117.7 ± 1.6 | 136.6 ± 1.3 | 133.1 ± 1.2 | 21.6 ± 2.4 | 114.1 ± 2.7 | 30.2 ± 2.4 | 58.7 ± 2.5 |
| Maze2D | Medium | 127.1 ± 1.4 | 141.3 ± 1.2 | 150.7 ± 1.0 | 151.4 ± 0.9 | 31.8 ± 1.7 | 97.8 ± 2.2 | 75.7 ± 2.1 | 153.4 ± 4.2 |
| Maze2D | Large | 168.2 ± 2.0 | 188.5 ± 1.6 | 203.6 ± 1.4 | 198.1 ± 1.5 | 15.1 ± 1.8 | 29.6 ± 1.9 | 174.6 ± 2.3 | 42.0 ± 2.4 |
| **Average** | | 142.4 | 149.2 | 163.6 | 160.9 | 22.8 | 80.5 | 93.5 | 84.7 |

Table 9 (part 2): Guided sampling algorithms **CG** and **None**.

| Dataset | Environment | CG Stride 1 | CG Stride 2/5 | CG Stride 4/15 | CG Stride 8/25 | None Stride 1 | None Stride 2/5 | None Stride 4/15 | None Stride 8/25 |
|---|---|---|---|---|---|---|---|---|---|
| Mixed | Kitchen | 64.0 ± 0.4 | 65.9 ± 0.4 | 61.3 ± 0.4 | — | 63.5 ± 0.4 | 63.5 ± 0.4 | 54.2 ± 0.3 | — |
| Partial | Kitchen | 58.9 ± 0.5 | 58.7 ± 0.6 | 58.4 ± 0.5 | — | 65.4 ± 0.5 | 56.0 ± 0.5 | 49.7 ± 0.5 | — |
| **Average** | | 61.5 | 62.3 | 59.9 | — | 64.5 | 59.8 | 52.0 | — |
| Antmaze-Large | Diverse | 79.1 ± 1.3 | 35.3 ± 1.5 | 20.6 ± 1.1 | 77.2 ± 1.3 | 73.1 ± 1.4 | 16.8 ± 1.6 | 11.6 ± 1.4 | 78.2 ± 1.8 |
| Antmaze-Large | Play | 64.9 ± 1.5 | 12.2 ± 0.9 | 8.6 ± 0.8 | 73.8 ± 1.4 | 58.6 ± 1.5 | 30.8 ± 2.0 | 0.2 ± 0.1 | 69.8 ± 2.0 |
| Antmaze-Medium | Diverse | 71.6 ± 1.4 | 57.3 ± 1.5 | 67.3 ± 1.4 | 59.0 ± 1.5 | 45.9 ± 2.2 | 57.0 ± 2.2 | 61.6 ± 2.1 | 55.0 ± 2.2 |
| Antmaze-Medium | Play | 50.6 ± 1.5 | 61.0 ± 1.5 | 78.3 ± 1.3 | 75.3 ± 1.4 | 60.8 ± 2.1 | 60.8 ± 2.1 | 67.0 ± 2.0 | 70.0 ± 2.0 |
| **Average** | | 66.6 | 41.5 | 43.7 | 71.3 | 59.6 | 41.4 | 35.1 | 68.3 |
| Maze2D | Umaze | 80.4 ± 2.4 | 64.3 ± 2.4 | 67.6 ± 2.4 | 81.0 ± 2.4 | 47.6 ± 2.4 | 38.9 ± 2.4 | 27.5 ± 2.4 | 35.3 ± 2.4 |
| Maze2D | Medium | 110.1 ± 2.2 | 113.3 ± 2.2 | 123.2 ± 2.1 | 126.9 ± 2.2 | 60.8 ± 2.3 | 61.9 ± 2.3 | 41.5 ± 2.2 | 47.3 ± 2.3 |
| Maze2D | Large | 148.6 ± 2.9 | 142.1 ± 2.9 | 131.4 ± 2.6 | 174.1 ± 2.6 | 68.1 ± 3.0 | 45.2 ± 2.8 | 42.6 ± 2.7 | 47.5 ± 2.6 |
| **Average** | | 113.0 | 106.6 | 107.4 | 127.3 | 58.8 | 48.7 | 37.2 | 43.4 |

Table 9: The effect of guided sampling algorithms for DV, with different planning strides. In "Stride x/y", x is for Kitchen, and y is for Maze2D & AntMaze. Data are Mean ± Standard Error over 500 episode seeds.

| Dataset | Environment | Transformer (DiT1D) | | | | U-Net (U-Net1D) | | | |
|---|---|---|---|---|---|---|---|---|---|
| | | Stride 1 | Stride 2/5 | Stride 4/15 | Stride 8/25 | Stride 1 | Stride 2/5 | Stride 4/15 | Stride 8/25 |
| Mixed | Kitchen | 67.2 ± 0.3 | 74.2 ± 0.1 | 73.6 ± 0.1 | — | 11.6 ± 0.5 | 35.2 ± 0.5 | 38.4 ± 0.8 | — |
| Partial | Kitchen | 87.8 ± 0.6 | 88.3 ± 0.4 | 94.0 ± 0.3 | — | 12.4 ± 0.5 | 28.8 ± 0.6 | 10.8 ± 0.5 | — |
| **Average** | | 77.5 | 81.3 | 83.8 | | 12.0 | 32.0 | 24.6 | |
| Antmaze-Large | Diverse | 72.4 ± 1.4 | 2.6 ± 0.1 | 71.2 ± 2.0 | 80.0 ± 1.8 | 53.7 ± 1.5 | 59.9 ± 1.5 | 76.7 ± 1.3 | 68.0 ± 1.4 |
| Antmaze-Large | Play | 61.2 ± 1.5 | 18.2 ± 0.5 | 76.2 ± 0.9 | 76.4 ± 2.0 | 53.4 ± 1.5 | 48.3 ± 1.5 | 80.8 ± 1.2 | 72.7 ± 1.4 |
| Antmaze-Medium | Diverse | 58.8 ± 1.5 | 24.2 ± 0.8 | 86.2 ± 1.6 | 87.4 ± 1.6 | 27.9 ± 1.4 | 55.5 ± 1.5 | 85.8 ± 1.1 | 81.1 ± 1.2 |
| Antmaze-Medium | Play | 38.8 ± 1.5 | 73.2 ± 1.5 | 83.8 ± 1.9 | 89.0 ± 1.6 | 28.5 ± 1.4 | 66.2 ± 1.4 | 81.0 ± 1.2 | 81.9 ± 1.2 |
| **Average** | | 57.8 | 29.6 | 79.4 | 83.2 | 40.9 | 57.5 | 81.1 | 75.9 |
| Maze2D | Large | 168.2 ± 2.0 | 188.5 ± 1.6 | 203.6 ± 1.4 | 198.1 ± 1.5 | 172.5 ± 1.9 | 168.7 ± 2.2 | 193.1 ± 1.5 | 189.7 ± 1.5 |
| Maze2D | Medium | 127.1 ± 1.4 | 141.3 ± 1.2 | 150.7 ± 1.0 | 151.4 ± 0.9 | 140.0 ± 1.1 | 138.7 ± 1.1 | 145.9 ± 1.1 | 146.6 ± 1.0 |
| Maze2D | Umaze | 131.8 ± 1.3 | 117.7 ± 1.6 | 136.6 ± 1.3 | 133.1 ± 1.2 | 126.4 ± 1.4 | 120.0 ± 1.6 | 126.5 ± 1.4 | 121.4 ± 1.4 |
| **Average** | | 142.4 | 149.2 | 163.6 | 160.9 | 146.3 | 142.5 | 155.2 | 152.6 |

Table 10: Changing denoising network backbone for DV, with different planning strides. In "Stride x/y", x is for Kitchen, and y is for Maze2D & AntMaze. Data are Mean ± Standard Error over 500 episode seeds.

| Dataset | Environment | Separate | | | | Joint | | | |
|---|---|---|---|---|---|---|---|---|---|
| | | Stride 1 | Stride 2/5 | Stride 4/15 | Stride 8/25 | Stride 1 | Stride 2/5 | Stride 4/15 | Stride 8/25 |
| Mixed | Kitchen | 67.2 ± 0.3 | 74.2 ± 0.1 | 73.6 ± 0.1 | — | 60.8 ± 0.4 | 62.9 ± 0.4 | 56.6 ± 0.6 | — |
| Partial | Kitchen | 87.8 ± 0.6 | 88.3 ± 0.4 | 94.0 ± 0.3 | — | 43.8 ± 0.7 | 53.7 ± 0.8 | 41.2 ± 0.7 | — |
| **Average** | | 77.5 | 81.3 | 83.8 | | 52.3 | 58.3 | 48.9 | |
| Antmaze-Large | Diverse | 72.4 ± 1.4 | 2.6 ± 0.1 | 71.2 ± 2.0 | 80.0 ± 1.8 | 49.9 ± 1.5 | 0.6 ± 0.2 | 0.2 ± 0.1 | 7.3 ± 0.8 |
| Antmaze-Large | Play | 61.2 ± 1.5 | 18.2 ± 0.5 | 76.2 ± 0.9 | 76.4 ± 2.0 | 56.7 ± 1.5 | 0.4 ± 0.2 | 0.1 ± 0.1 | 3.7 ± 0.5 |
| Antmaze-Medium | Diverse | 58.8 ± 1.5 | 24.2 ± 0.8 | 86.2 ± 1.6 | 87.4 ± 1.6 | 36.3 ± 1.5 | 2.5 ± 0.4 | 0.3 ± 0.1 | 6.3 ± 0.7 |
| Antmaze-Medium | Play | 38.8 ± 1.5 | 73.2 ± 1.5 | 83.8 ± 1.9 | 89.0 ± 1.6 | 34.0 ± 1.4 | 2.7 ± 0.5 | 3.7 ± 0.6 | 27.8 ± 1.4 |
| **Average** | | 57.8 | 29.6 | 79.4 | 83.2 | 44.2 | 1.6 | 1.1 | 11.3 |
| Maze2D | Large | 168.2 ± 2.0 | 188.5 ± 1.6 | 203.6 ± 1.4 | 198.1 ± 1.5 | 187.0 ± 1.6 | 202.6 ± 1.4 | 202.9 ± 1.5 | 198.5 ± 1.6 |
| Maze2D | Medium | 127.1 ± 1.4 | 141.3 ± 1.2 | 150.7 ± 1.0 | 151.4 ± 0.9 | 141.4 ± 1.1 | 143.7 ± 1.1 | 156.0 ± 0.9 | 153.8 ± 0.9 |
| Maze2D | Umaze | 131.8 ± 1.3 | 117.7 ± 1.6 | 136.6 ± 1.3 | 133.1 ± 1.2 | 124.9 ± 1.6 | 139.8 ± 1.2 | 146.3 ± 1.1 | 141.1 ± 1.2 |
| **Average** | | 142.4 | 149.2 | 163.6 | 160.9 | 151.1 | 162.0 | 168.4 | 164.5 |

Table 11: The impact of action generation method for DV, with different planning strides. In "Stride x/y", x is for Kitchen, and y is for Maze2D & AntMaze. Data are Mean ± Standard Error over 500 episode seeds.

| | | IL | None-diffusion Policies | | | Diffusion Policies | | | | | | Diffusion Planners | | | | |
|---|---|---|---|---|---|---|---|---|---|---|---|---|---|---|---|---|
| Dataset | Environment | BC | BCQ | CQL | IQL | SfBC | DQL | DQL* | IDQL | IDQL* | CEP | Diffuser | AdaptDiffuser | DD | HD | DV |
| Medium-Expert | HalfCheetah | 35.8 | 64.7 | 62.4 | 86.7 | 92.6±0.5 | 96.8±0.3 | 95.5±0.1 | 95.9 | 91.3±0.6 | 93.5±0.3 | 88.9±0.3 | 89.6±0.8 | 90.6±1.3 | 92.5±0.3 | 92.7±0.3 |
| Medium-Replay | HalfCheetah | 38.4 | 38.2 | 46.2 | 44.2 | 37.1±1.7 | 47.8±0.3 | 47.9±0.0 | 45.9 | 46.5±0.3 | 47.6±1.4 | 37.7±0.5 | 38.3±0.9 | 39.3±4.1 | 38.1±0.7 | 45.8±0.1 |
| Medium | HalfCheetah | 36.1 | 44.4 | 44.4 | 47.4 | 45.9±2.2 | 51.1±0.5 | 52.3±0.2 | 51.0 | 51.5±0.1 | 54.1±0.4 | 42.8±0.3 | 44.2±0.6 | 49.1±1.0 | 46.7±0.2 | 50.4±0.0 |
| Medium-Expert | Hopper | 111.9 | 110.9 | 98.7 | 91.5 | 108.6±2.1 | 111.1±1.3 | 111.1±0.4 | 108.6 | 110.1±0.7 | 108.0±2.5 | 103.3±1.3 | 111.6±2.0 | 111.8±1.8 | 115.3±1.1 | 110.0±0.5 |
| Medium-Replay | Hopper | 11.3 | 33.1 | 48.6 | 94.7 | 86.2±9.1 | 101.3±0.6 | 101.6±0.0 | 92.1 | 99.4±0.1 | 96.9±2.6 | 93.6±0.4 | 92.2±1.5 | 100.0±0.7 | 99.3±0.3 | 91.9±0.0 |
| Medium | Hopper | 29.0 | 54.5 | 58.0 | 66.3 | 57.1±4.1 | 90.5±4.6 | 90.5±1.3 | 65.4 | 70.1±2.0 | 98.0±2.6 | 74.3±1.4 | 96.6±2.7 | 79.3±3.6 | 84.0±0.6 | 83.6±1.2 |
| Medium-Expert | Walker2d | 6.4 | 57.5 | 111.0 | 109.6 | 109.8±0.2 | 110.1±0.3 | 111.6±0.0 | 112.7 | 110.6±0.0 | 110.7±0.6 | 106.9±0.2 | 108.2±0.8 | 108.8±1.7 | 107.1±1.1 | 109.2±0.0 |
| Medium-Replay | Walker2d | 11.8 | 15.0 | 26.7 | 73.9 | 65.1±5.6 | 95.5±1.5 | 98.2±0.1 | 85.1 | 89.1±2.4 | 84.4±4.1 | 70.6±1.6 | 84.7±3.1 | 75.0±4.3 | 94.7±0.7 | 85.0±0.5 |
| Medium | Walker2d | 6.6 | 53.1 | 79.2 | 78.3 | 77.9±2.5 | 87.0±0.9 | 86.8±0.2 | 82.5 | 88.1±0.4 | 86.0±0.7 | 79.6±0.6 | 84.4±2.6 | 82.5±1.4 | 84.1±2.2 | 82.8±0.1 |
| **Average** | | 31.9 | 52.0 | 63.9 | 77.0 | 75.6 | 87.9 | **89.1** | 82.1 | 84.1 | 86.6 | 77.5 | 83.3 | 81.8 | 84.6 | 83.5 |
| Mixed | Kitchen | 47.5 | 8.1 | 51.0 | 51.0 | 45.4±1.6 | 62.6±5.1 | 55.1±1.58 | 66.5 | 66.5±4.1 | — | 52.5±2.5 | 51.8±0.8 | 75.0±0.0 | 71.7±2.7 | 73.6±0.1 |
| Partial | Kitchen | 33.8 | 18.9 | 49.8 | 46.3 | 47.9±4.1 | 60.5±6.9 | 65.5±1.38 | 66.7 | 66.7±2.5 | — | 55.7±1.3 | 55.5±0.4 | 56.5±5.8 | 73.3±1.4 | 94.0±0.3 |
| **Average** | | 40.7 | 13.5 | 50.4 | 48.7 | 46.7 | 61.6 | 60.3 | 66.6 | 66.6 | — | 54.1 | 53.7 | 65.8 | 72.5 | 83.8 |
| Diverse | Antmaze-Large | 0.0 | 2.2 | 15.8 | 47.5 | 45.5±6.6 | 56.6±7.6 | 70.6±3.7 | 67.9 | 40.0±11.4 | 64.8±5.5 | 27.3±2.4 | 8.7±2.5 | 0.0±0.0 | 83.6±5.8 | 80.0±1.8 |
| Play | Antmaze-Large | 0.0 | 6.7 | 14.9 | 39.6 | 59.3±14.3 | 46.4±8.3 | 81.3±3.1 | 63.5 | 48.7±4.7 | 66.6±9.8 | 17.3±1.9 | 5.3±3.4 | 0.0±0.0 | — | 76.4±2.0 |
| Diverse | Antmaze-Medium | 0.0 | 0.0 | 53.7 | 70.0 | 81.3±2.6 | 78.6±10.3 | 82.6±3.0 | 84.8 | 83.3±5.0 | 83.8±3.5 | 2.0±1.6 | 6.0±3.3 | 4.0±2.8 | 88.7±8.1 | 87.4±1.6 |
| Play | Antmaze-Medium | 0.0 | 0.0 | 61.2 | 71.2 | 82.0±3.1 | 76.6±10.8 | 87.3±2.7 | 84.5 | 67.3±5.7 | 83.6±4.4 | 6.7±5.7 | 12.0±7.5 | 8.0±4.3 | — | 89.0±1.6 |
| **Average** | | 0.0 | 2.2 | 36.4 | 57.1 | 67.0 | 64.6 | 80.5 | 75.2 | 59.8 | 74.7 | 13.3 | 8.0 | 3.0 | — | 83.2 |
| Large | Maze2D | 5 | 6.2 | 12.5 | 58.6 | 74.4±1.7 | 186.8±1.7 | — | 90.1 | — | — | 123 | 167.9±5.0 | — | 128.4±3.6 | 203.6±1.4 |
| Medium | Maze2D | 30.3 | 8.3 | 5.0 | 34.9 | 73.8±2.9 | 152.0±0.8 | — | 89.5 | — | — | 121.5 | 129.9±4.6 | — | 135.6±3.0 | 150.7±1.0 |
| Umaze | Maze2D | 3.8 | 12.8 | 5.7 | 47.4 | 73.9±6.6 | 140.6±1.0 | — | 57.9 | — | — | 113.9 | 135.1±5.8 | — | 155.8±2.5 | 136.6±1.3 |
| **Average** | | 13.0 | 9.1 | 7.7 | 47.0 | 74.0 | 159.8 | — | 79.2 | — | — | 119.5 | 144.3 | — | 139.9 | **163.6** |

Table 12: Normalized performance of various offline-RL methods. Data are Mean ± Standard Error (if available).

