# OpenReview forum: "What Makes a Good Diffusion Planner for Decision Making?"
_ICLR.cc/2025/Conference — ICLR 2025 Spotlight_

### Official Review · Reviewer_fADJ · 2024-10-21

**Soundness:** 2
**Presentation:** 3
**Contribution:** 3
**Rating:** 8
**Confidence:** 3

**Summary:**

In this paper, the authors investigated the design choices for diffusion model-based offline RL methods.
The design choices mainly focused on planning strategy, network architecture, guided sampling, and action generation (whether to generate both state and action directly, or generate only the state and estimate the action separately using an inverse dynamics model).
The tasks used in the study were Maze2D, AntMaze, and Franka kitchen (MuJoCo locomotion also used in section 4.6).

**Strengths:**

This paper investigates the design choices for diffusion model-based offline RL methods and identifies effective components. Although various (algorithm/implmentation) designs have been proposed in previous research on diffusion model-based offline RL methods, their effectiveness in a unified framework has not been sufficiently explored.
In general, the performance of reinforcement learning methods largely depends on design choices.
Therefore, this paper, which provides insights into effective design choices, has value on the engineering front.

**Weaknesses:**

The number of tasks used to investigate the design choices is limited. This paper focuses on Maze2D (2 tasks), AntMaze (3 tasks), and Franka kitchen (4 tasks) (with MuJoCo locomotion tasks also included in section 4.6). However, for a paper investigating design choices, this is fewer than the number of tasks typically covered in papers accepted at ICLR (or conferences of a similar level). For instance, the paper [1] that investigated the implementation design of Offline + Online RL used 30 tasks in its study.

Moreover, the paper does not verify the insights/findings on a different set of tasks (i.e., validation tasks) from those used in the design choice investigation. This leaves uncertainty about how generalizable the insights are (or whether they are simply overfitted to the tasks examined).


[1] Ball, Philip J., et al. "Efficient online reinforcement learning with offline data." International Conference on Machine Learning. PMLR, 2023.


Minor comments:

Line 018:
> We trained and evaluated over 6,000 diffusion models

I didn’t quite understand the breakdown of these 6,000 diffusion models. Were most of these models the ones trained and evaluated through the grid search mentioned in the step (1) in Section 3.2?

Line 174:
> (1) Conduct a comprehensive search on the key components (Sect. 3.1) by combining grid search and manual tuning to obtain the best results.

What exactly does "manual tuning" refer to in this context?

Figure 5.
It seems that the Transformer score for Kitchen-M doesn’t have a confidence interval.
Also, I wasn’t clear on what the confidence intervals in the other parts of this figure represent (are they calculated based on 500 episode seeds?).

Typoes:
line 101: Zhang et al., 2022) In  -> Zhang et al., 2022). In
line 158:  Chen et al., 2024)) -> Chen et al., 2024).
line 479: planning(Sect. 4.6) -> planning (Sect. 4.6).

**Questions:**

Please refer to my previous comment on the weaknesses.
If either (1) validation results from tasks other than those used to investigate the design choices, or (2) validation results from 20-30 tasks were provided to support the insights on design choices, I would be inclined to recommend an Accept (assuming other reviewers do not point out any major weaknesses that I may have overlooked).

---

> ### Author Response · Authors · 2024-11-21
> **Response to reviewer fADJ**
>
> > W1. The number of tasks used to investigate the design choices is limited.
> >
> > W2. The paper does not verify the insights/findings on a different set of tasks (i.e., validation tasks) from those used in the design choice investigation ... leaves uncertainty about how generalizable the insights are.
>
> Thanks for the suggestion. We have conducted experiments on additional datasets (Adroit, see **Sect. 4.7** and **Appendix C**). The insights and conclusions drawn from the new results are actually consistent with the existing ones (i.e., the takeaways in **Sect. 4.8**). We have also included the code in the supplementary material to ensure reproducibility.
>
> In particular, we observed that the following **key findings are consistent**. For Adroit tasks, the empirical results (**Sect. 4.7** and **Appendix C**) show:
> - It is better to generate a state sequence and then compute the action using an inverse dynamics model than to jointly generate state and action sequences, consistent with **Sect. 4.1** (action generation).
> - Jump-step planning is slightly better than dense-step planning, consistent with **Sect. 4.2** (planning strategy).
> - Transformer outperforms UNet as the backbone for the denoising network, consistent with **Sect. 4.3** (denoising network backbone).
> - A one-layer Transformer performs poorly, while there is no improvement with more than two layers, consistent with **Sect. 4.4** (impact of network size).
> - MCSS outperforms CFG and CG, consistent with **Sect. 4.5** (guidance sampling algorithms).
>
> > W3. I didn’t quite understand the breakdown of these 6,000 diffusion models. Were most of these models the ones trained and evaluated through the grid search?
> >
> > W4. What exactly does "manual tuning" refer to in this context?
>
> We apologize for the lack of clarity regarding our hyperparameter search process. Training diffusion models demands significantly more computational resources compared to traditional Gaussian policies, making exhaustive grid searches impractical. Instead, we conducted several rounds of hyperparameter tuning, where each round focused on a subset of hyperparameters that we identified as most influential based on prior experiments and domain knowledge. The “manual tuning” refers to this iterative process of selecting which hyperparameters to explore in each round, guided by preliminary results and insights.
>
> The "6,000+ models" is the cumulative number of experiments executed on our computing cluster throughout this iterative tuning process. These experiments include various combinations of hyperparameters tested across different rounds of searches. We have now included explanations of our hyperparameter tuning strategy in the revised manuscript (**Appendix B.4**) to provide better transparency.
>
> > W5. It seems that the Transformer score for Kitchen-M doesn't have a confidence interval.
> >
> > W6. I wasn't clear on what the confidence intervals in the other parts of this figure represent (are they calculated based on 500 episode seeds?).
>
> We appreciate your careful observation. In fact, the confidence interval for Kitchen-M exists, but it is very small and not visible on the plot. We have added an explanation in the caption of **Fig. 5** to clarify this issue. Additionally, we have provided the numerical results in **Table 5** in the Appendix.
>
> Yes. These confidence intervals are calculated based on 500 episode seeds, representing standard errors.
>
> > Typoes:
> line 101: Zhang et al., 2022) In -> Zhang et al., 2022). In
> line 158: Chen et al., 2024)) -> Chen et al., 2024).
> line 479: planning(Sect. 4.6) -> planning (Sect. 4.6).
>
> Thank you for your careful reading. We have fixed these typos.
>
> Finally, we would like to thank you again for the useful comments, which we believe have significantly enhanced our work. If you have any further comments, we would be glad to have a deeper discussion with you.

---

> > ### Comment · Reviewer_fADJ · 2024-11-22
> > **Response to the authors**
> >
> > Thank you for your response.
> > As you’ve addressed my concerns, I will raise my recommendation to Accept.

---

> > > ### Author Response · Authors · 2024-11-25
> > >
> > > Thank you for letting us know that we have addressed your concerns. Your feedback has greatly improved our work!

---

### Official Review · Reviewer_ro9k · 2024-10-23

**Soundness:** 3
**Presentation:** 3
**Contribution:** 2
**Rating:** 6
**Confidence:** 3

**Summary:**

The paper explores the design choices in diffusion model planning within offline reinforcement learning (RL). Through experiments on over 6,000 models, the paper systematically investigates key components of diffusion planning, including sampling algorithms, network architectures, action generation methods, and planning strategies. The study finds that some design choices, such as unconditional sampling outperforming guided sampling and Transformer outperforming U-Net, lead to better performance. Based on these insights, the paper proposes a simple yet strong baseline model called Diffusion Veteran (DV), which achieves state-of-the-art results on standard offline RL benchmarks.

**Strengths:**

1.Comprehensive empirical study: The paper conducts a large-scale experimental study, using controlled variable methods to analyze the impact of each component on model performance, providing rich data support.
2.Innovative insights: The study reveals design choices that contrast with common practices in diffusion planning, such as the advantages of unconditional sampling and the use of Transformer, offering new directions for future research.
3.Simple yet effective baseline model: The proposed DV model is simple but performs exceptionally well, demonstrating high generalizability and effectiveness, laying a solid foundation for further research.
4.Wide applicability: The DV model performs well in multiple tasks such as maze navigation and robot manipulation, demonstrating its adaptability and broad applicability.

**Weaknesses:**

1.Limited exploration of long-term dependencies: While the paper discusses the importance of handling long-term dependencies using Transformer, it does not delve deeply into how this is manifested across different tasks. The related discussion could be more robust.
2.Potential typo in Equation 2.1: There seems to be a typo on the right-hand side of Equation 2.1, where S(t−1) appears, which might be incorrect.

**Questions:**

1.You mention that unconditional sampling outperforms guided sampling, which contrasts with results in typical image generation tasks. Could you elaborate on the underlying reasons behind this phenomenon?
2.The paper primarily focuses on state-based tasks. Are there plans to extend the study to vision-based or goal-conditioned reinforcement learning tasks?

---

> ### Author Response · Authors · 2024-11-21
> **Response to reviewer ro9k**
>
> > W1. Limited exploration of long-term dependencies: While the paper discusses the importance of handling long-term dependencies using Transformer, it does not delve deeply into how this is manifested across different tasks. The related discussion could be more robust.
>
> Thank you for helping us strengthen our experiments. We have provided the attention maps for various tasks in **Appendix D.1**. Although the attention patterns vary across different tasks, they all exhibit long-term attention, suggesting that long-term dependencies are common among these tasks and explaining why Transformers outperform UNet. The attention patterns typically feature slashes, which attend to a fixed number of steps prior, and vertical lines, which attend to key steps. We have complemented **Sect. 4.3** with these additional analyses.
>
> > W2. Potential typo in Equation 2.1: There seems to be a typo on the right-hand side of Equation 2.1, where S(t−1) appears, which might be incorrect.
>
> Thank you for pointing out this typo. It should indeed be **S(t+1)**, and we have fixed it.
>
> > Q1. You mention that unconditional sampling outperforms guided sampling, which contrasts with results in typical image generation tasks. Could you elaborate on the underlying reasons behind this phenomenon?
>
> This is indeed an interesting question that warrants deeper investigation. In addition to the analysis provided in the manuscript (**Sect. 4.5**), we elaborate on the potential reasons from the following perspective:
>
> **Theorem 1**: Suppose the reward distribution of the trajectories generated by unconditional models is identical to the reward distribution of the trajectories in the training dataset. Assume that, in the training dataset, the proportion of trajectories with a reward above $R$ is $p$. If we want MCSS's reward to exceed $R$, the expected number of sampling times is $\frac{1}{p}$.
>
> **Proof**: We apply the total expectation formula to compute the expected value, denoted as $E$. For the first generated trajectory, if the reward is above $R$, the expected value is $1$. Otherwise, the expected value is $1 + E$. We then have $p + (1-p) \cdot (1 + E) = E$, which simplifies to $E = \frac{1}{p}$.
>
> Using Theorem 1, we evaluated when MCSS can outperform guidance. We set the threshold $R = 0.9R_{\max}$. The expected number of sampling times for kitchen, antmaze, and maze2d are $66.66$, $6.45$, and $13.51$ (where $p$ can be obtained from the data in Fig. 7b), respectively. In the experiments, we set the number of sampling times to $50$. As a result, MCSS achieves better performance than diffusion guidance in antmaze and maze2d, but its performance is slightly inferior in the kitchen scenario.
>
> > Q2. The paper primarily focuses on state-based tasks. Are there plans to extend the study to vision-based or goal-conditioned reinforcement learning tasks?
>
> This is a great point! Yes, we plan to conduct future studies on vision-based and goal-conditioned reinforcement learning (RL) tasks for real-world applications. We believe that combining proprioception (joint states), vision, and possibly force feedback/tactile sensing, along with goal-directed planning, will result in more powerful robotic AI.
>
> Thank you again for your constructive feedback! We hope the responses above have addressed all your comments. Please kindly let us know if you have additional suggestions, and we would be more than happy to discuss them.

---

> > ### Comment · Reviewer_ro9k · 2024-11-23
> >
> > The author's rebuttal effectively addressed my concerns, and I improved my score

---

> > > ### Author Response · Authors · 2024-11-25
> > >
> > > Thank you for letting us know that we have addressed your concerns. Your feedback has greatly improved our work!

---

### Official Review · Reviewer_7fqj · 2024-10-28

**Soundness:** 4
**Presentation:** 4
**Contribution:** 4
**Rating:** 10
**Confidence:** 4

**Summary:**

This paper presents an extensive experimental study aimed at understanding the factors that contribute to an effective diffusion planner for decision-making in offline reinforcement learning. The authors provide valuable insights into the role of various components within diffusion models. Building on these insights, they propose a straightforward yet robust diffusion planning approach that delivers state-of-the-art (SOTA) performance in standard offline RL benchmarks.

**Strengths:**

1. This paper is well-organized and easy to follow.
2. The empirical analysis is comprehensive, providing solid support for the conclusions.
3. Each conclusion is accompanied by decent explanations

**Weaknesses:**

While the paper provides strong evidence for the effectiveness of the proposed methods on the D4RL dataset, it is unclear how generalizable these findings are to other types of decision-making problems or datasets. More diverse datasets could strengthen the claims.

**Questions:**

No

---

> ### Author Response · Authors · 2024-11-21
> **Response to reviewer 7fqj**
>
> Thank you very much for reviewing our work and providing your thoughts.
>
> > While the paper provides strong evidence for the effectiveness of the proposed methods on the D4RL dataset, it is unclear how generalizable these findings are to other types of decision-making problems or datasets. More diverse datasets could strengthen the claims.
>
> We appreciate your suggestion and agree that our work can be strengthened by incorporating more datasets. To address this, we conducted experiments using our model on additional datasets (Adroit, including 8 subtasks) to validate our findings, which have been discussed in **Sect. 4.7** of the revised manuscript (detailed in **Appendix C**). For the experiments on Adroit, we inherited the hyperparameters used for Kitchen. The source code is included in the supplementary material and will also be published.
>
> We observed that the following **key findings are consistent**. For Adroit tasks, the empirical results (Sect. 4.7 and Appendix C) show:
> - Generating a state sequence and then computing the action using an inverse dynamics model is better than jointly generating state and action sequences, consistent with Sect. 4.1 (action generation).
> - Jump-step planning is slightly better than dense-step planning, consistent with Sect. 4.2 (planning strategy).
> - Transformer outperforms UNet as the backbone for the denoising network, consistent with Sect. 4.3 (denoising network backbone).
> - A one-layer Transformer performs poorly, while there is no improvement with more than two layers, consistent with Sect. 4.4 (impact of network size).
> - MCSS outperforms CFG and CG, consistent with Sect. 4.5 (guidance sampling algorithms).
>
> We believe the above addresses your concern. Thank you for helping to improve our work. Should you have any other comments, please kindly let us know.

---

### Official Review · Reviewer_u6s9 · 2024-11-01

**Soundness:** 2
**Presentation:** 3
**Contribution:** 2
**Rating:** 6
**Confidence:** 2

**Summary:**

This paper analyses key components (guided sampling algorithms, network architectures, action generation methods, and planning strategies) critical to decision-making in diffusion planning.  The paper gives practical tips about the choices
and provides insights into the strengths and limitations of diffusion planning. The experiments in the paper are very comprehensive.

**Strengths:**

The experiments in the paper are very comprehensive.

**Weaknesses:**

Although the experiments in the paper are rich, readers still want to see how the original innovation in theory can better apply diffusion models to decision-making tasks

**Questions:**

No

---

> ### Author Response · Authors · 2024-11-21
> **Response to reviewer u6s9**
>
> Thank you for reviewing our paper and raising the concern regarding "original innovation in theory." On one hand, we would like to emphasize that the controlled experiments presented in this paper have the potential to inspire future theoretical innovations in the field. For instance, one could develop a theory to provide insights into the surprisingly good performance of MCSS, as discussed in Section 4.5.
>
> **Theorem 1**: Suppose the reward distribution of the trajectories generated by unconditional models is identical to the reward distribution of the trajectories in the training dataset. Assume that, in the training dataset, the proportion of trajectories with a reward above $R$ is $p$. If we want MCSS's reward to exceed $R$, the expected number of sampling times is $\frac{1}{p}$.
>
> **Proof**: We apply the total expectation formula to compute the expected value, denoted as $E$. For the first generated trajectory, if the reward is above $R$, the expected value is $1$. Otherwise, the expected value is $1 + E$. We then have $p + (1-p) \cdot (1 + E) = E$, which simplifies to $E = \frac{1}{p}$.
>
> Using Theorem 1, we evaluate when MCSS can outperform guidance. We set the threshold $R = 0.9R_{\max}$. The expected number of sampling times for kitchen, antmaze, and maze2d are $66.66$, $6.45$, and $13.51$ (where $p$ can be obtained from the data in Fig. 7b), respectively. In the experiments, we set the number of sampling times to $50$. As a result, MCSS achieves better performance than diffusion guidance in antmaze and maze2d, but its performance is slightly inferior in the kitchen scenario.
>
> A similar theory could be developed to provide insights into the experiments in other subsections, which could lead to further performance enhancements in this field. However, complete theoretical justifications for the experiments are beyond the scope of this paper and are left for future work. Nonetheless, we believe our empirical results pave the way for future theoretical innovations in the field.
>
> We hope the above addresses your concerns. Moving forward, we will carefully consider how to draw insights from theory to further strengthen our understanding of diffusion planning and decision-making.

---

### Author Response · Authors · 2024-11-21
**Response to all reviewers**

We sincerely thank all the reviewers for reading our manuscript and providing insightful advice to improve it. Based on the constructive feedback, we have carefully revised the manuscript. The reviewers' comments have significantly contributed to enhancing the quality of the paper. In the revised manuscript, we have mainly made the following changes:

1. We included more tasks to strengthen our claims. Specifically, we conducted validation experiments on **eight new tasks from the Adroit Hand environment**. Our findings confirm that the conclusions are consistent with the new results (see Sect. 4.7 and Appendix C in the revised paper). We have also provided the corresponding source code in the supplementary material to ensure reproducibility.
2. We added further discussion about the presence of long-term dependency reflected by the attention weights in Transformers in Sect. 4.3 (and Appendix D.1).

The main changes in the manuscript are highlighted in blue.

By incorporating the reviewers' comments, the main contributions of our work are now more clearly substantiated with robust experimental results. We believe that this current work takes an intial step towards systematically understanding and applying diffusion models for decision making. We hope the extensive experiments presented in this paper will inspire future theoretical analyses and algorithm development. Please feel free to share any additional comments on the manuscript or the changes.

---

### Author Response · Authors · 2024-12-03
**Final Remarks**

We greatly appreciate reviewers 7fqj, fADJ, ro9k, u6s9 as well as AC/SAC/PC for the dedication to the review process. As today is the final day of the rebuttal period, we would like to highlight the following points to further clarify the contributions of our work:

- While the original Diffuser paper (Janner et al., 2022) answered **whether** diffusion models can be used for planning, we addressed **how** to unleash the potential of diffusion planners. We have identified several design choices that, contrary to common practices in the literature, surprisingly enhance performance.

- Our proposed DV model achieved new SOTA results on three task sets (Kitchen, AntMaze, Maze2D) within the standard offline RL benchmark D4RL, thereby providing **a simple yet strong baseline** for future studies.

- Our empirical analysis dissects the components of diffusion planners. The insights gained are valuable in two ways: (1) they can **greatly reduce engineering efforts** in future experimental work, and (2) they provide **solid experimental evidence** to inform future theoretical research.

If you have **any remaining questions, please kindly let us know by the due**. Thank you once again for your valuable feedback throughout this process.

---

### Meta-Review · Area_Chair_8B2W · 2024-12-16

**Metareview:**

The paper has been well-received, with all reviewers praising its comprehensive empirical analysis, innovative insights, and the simplicity and effectiveness of the proposed DV model. The reviewers agree that the paper makes a significant contribution to the field of offline reinforcement learning by providing valuable insights into the design choices for diffusion models.

Strengths
-----------

- **Comprehensive empirical analysis:** The reviewers recognise the thoroughness of the experimental study, which uses controlled variables to analyze the impact of different components on model performance. This rigorous approach provides strong evidence for the authors' claims.

- **Innovative insights:** The paper challenges common practices in diffusion planning by demonstrating the advantages of unconditional sampling and the use of Transformer. These findings offer new directions for future research in the field.

- **Simple and effective model:** The proposed DV model is simple and effective, with strong performance across different tasks, including maze navigation and robot manipulation, suggesting high generalizability and effectiveness.

- **Clear presentation:** The paper is well-organized and easy to follow, with clear explanations for each conclusion.

Weaknesses
--------------
- **Limited task diversity:** A major concern raised by the reviewers is the limited number and type of tasks used to evaluate the proposed method. They suggest that including more diverse datasets and validation tasks would strengthen the paper's claims and demonstrate the generalizability of the findings.

- **Long-term dependencies:** While the paper discusses the importance of handling long-term dependencies, one reviewer feels that the discussion could be more in-depth, particularly regarding how this is manifested across different tasks.

- **Minor issues:** Reviewers pointed out a few minor issues like potential typos in equations and unclear explanations of confidence intervals in figures.

Based on the reviewers' feedback, I recommend the Authors consider including a discussion about long-term dependencies in the final version. For instance, providing a detailed analysis of how the model handles long-term dependencies in different tasks. Finally, I recommend addressing minor issues highlighted by Reviewers.

**Additional Comments On Reviewer Discussion:**

The rebuttal has been crucial to address concerns about how the proposed method handle long-term dependencies and to show additional empirical results. This resulted in the paper achieving unanimously high scores.

---

### Decision · Program_Chairs · 2025-01-22

Accept (Spotlight)